# Chromatin accessibility landscape of relapsed pediatric B-lineage acute lymphoblastic leukemia

Han Wang[1,4], Huiying Sun[1,4], Bilin Liang[1], Fang Zhang[2], Fan Yang[1], Bowen Cui [1], Lixia Ding[2], Xiang Wang[2], Ronghua Wang[1], Jiaoyang Cai[2], Yanjing Tang[2], Jianan Rao[1], Wenting Hu[2], Shuang Zhao[1], Wenyan Wu[1], Xiaoxiao Chen[2], Kefei Wu [2], Junchen Lai[2], Yangyang Xie[2], Benshang Li[2], Jingyan Tang[2], Shuhong Shen [2,3] ✉ & Yu Liu [1,2,3] ✉

For around half of the pediatric B-lineage acute lymphoblastic leukemia (B-ALL) patients, the molecular mechanism of relapse remains unclear. To fill this gap in knowledge, here we characterize the chromatin accessibility landscape in pediatric relapsed B-ALL. We observe rewired accessible chromatin regions (ACRs) associated with transcription dysregulation in leukemia cells as compared with normal B-cell progenitors. We show that over a quarter of the ACRs in B-ALL are in quiescent regions with high heterogeneity among B-ALLs. We identify subtype-specific and allele-imbalanced chromatin accessibility by integrating multi-omics data. By characterizing the differential ACRs between diagnosis and relapse in B-ALL, we identify alterations in chromatin accessibility during drug treatment. Further analysis of ACRs associated with relapse free survival leads to the identification of a subgroup of B-ALL which show early relapse. These data provide an advanced and integrative portrait of the importance of chromatin accessibility alterations in tumorigenesis and drug responses.

Acute lymphoblastic leukemia (ALL) is the most common childhood cancer. B-lineage ALL (B-ALL) accounts for about 80% of pediatric ALL cases. Genomic analyses of large cohorts have identified more than 20 B-ALL subtypes with distinct genetic alterations[1], which has enabled risk stratification and precision treatment. This, in combination with other treatment advances, has increased the patient survival rate to over 90%[2]. However, patients with refractory and relapsed B-ALL show a dismal prognosis, with 5-year survival rate <50%[3,4]. Genomic analyses of relapsed ALL patients have revealed several somatic mutations acquired during chemotherapy that could cause drug resistance of leukemia cells. These include mutations in *NT5C2*[5,6], which increases

cell resistance to purine analogs, *PRPS1/PRPS2*[7], *FPGS*[8], *NR3C1/NR3C2*[9], and *CREBBP*[10], among others. However, these genomic aberrations could only be detected in a subset of relapsed tumors, and the mechanisms of drug resistance and relapse remain unknown for nearly half of such patients. Moreover, most of these studies focused on analysis of coding genes in the genome, leaving the noncoding genomic counterpart largely unexplored.

Epigenomics analysis is one important way to interpret the function of the noncoding genome. Recently studies have unveiled epigenomics features as an essential characterization of tumor cells, with implications in pathogenesis, clinical behavior, and therapy[11,12]. Among

[1]Pediatric Translational Medicine Institute, Shanghai Children's Medical Center, School of Medicine, Shanghai Jiao Tong University, Shanghai, China. [2]Key Laboratory of Pediatric Hematology and Oncology Ministry of Health, Department of Hematology and Oncology, Shanghai Children's Medical Center, School of Medicine, Shanghai Jiao Tong University, Shanghai, China. [3]Fujian Children's Hospital, Fujian Branch of Shanghai Children's Medical Center Affiliated to Shanghai Jiao Tong University School of Medicine, Fuzhou, China. [4]These authors contributed equally: Han Wang, Huiying Sun. ✉e-mail: shenshuhong@scmc.com.cn; yu.liu@sjtu.edu.cn

all epigenomic marks, histone modifications and DNA methylation have been the most widely studied to gain insight into epigenomic dysregulation[13,14]. Chromatin accessibility is a hallmark of DNA regulatory elements[15], and emerging evidence shows that it plays a significant role in cancer[16,17]. The advent and optimization of the assay for transposase-accessible chromatin using sequencing (ATAC-seq) have made it possible to profile chromatin accessibility genome-wide in primary cancers[18,19]. Using this technology, a recent study showed that lymphocyte-specific open chromatin regions pre-determine glucocorticoid resistance in ALL[20], suggesting the potential role of chromatin accessibility features in B-ALL drug resistance and relapse. However, knowledge about the chromatin accessibility profiles in primary pediatric B-ALL and the changes in accessibility that occur during relapse is still lacking.

In this study, we present the chromatin accessibility profiles of 61 relapsed pediatric B-ALL patients. The chromatin accessibility features are interpreted by incorporating multiple genome-wide sequencing data, namely whole genome sequencing (WGS), transcriptome sequencing (RNA-seq), and chromatin immunoprecipitation sequencing (ChIP-seq) with antibodies against H3K27ac, which is an indicator of active enhancers. By comparing with B-cell progenitors, we show the rewired chromatin accessibility in B-ALL, which is associated with leukemogenesis. Further comparison between diagnosis and relapse unveils alterations in chromatin accessibility in response to drug treatment in B-ALL. Moreover, a chromatin-accessible signature is identified distinguishing B-ALL patients with inferior prognoses.

## Results

### Chromatin accessibility landscape of pediatric B-ALL

A total of 144 chromatin accessibility profiles were generated from 79 pediatric B-ALL tumors collected from 61 relapsed B-ALL patients treated at Shanghai Children's Medical Center (Supplementary Data 1). Multiple genomics sequencing data were also generated or available from a previous study[5], namely WGS data for the diagnosis-remission-relapse trios from 32 patients, RNA-seq data for 89 tumors derived from 57 B-ALL patients, and H3K27ac ChIP-seq data for 12 tumors from 11 B-ALL patients (Fig. 1a and Supplementary Fig. 1a). The molecular subtype for each B-ALL patient was determined by integrating the driver genomic translocations from WGS, fusions and gene expression signatures from RNA-seq, and karyotype and FISH results from clinical testing (Methods and Supplementary Data 1). The following 11 B-ALL subtypes were included in this analysis, namely hyperdiploidy ($n = 20$), ETV6::RUNX1 ($n = 11$), TCF3::PBX1 ($n = 5$), KMT2A rearranged ($n = 5$), BCR::ABL1 ($n = 3$), BCR::ABL1-like ($n = 4$), ZNF384 ($n = 3$), PAX5alt ($n = 2$), TCF3::HLF ($n = 1$), hypodiploidy ($n = 1$), MEF2D ($n = 1$), and five cases with unclassified subtype, which were designated B-other. Living tumor cells were purified with flow cytometry against tumor-specific antigens to reduce the noise from normal cells (Methods, Supplementary Fig. 1b and Supplementary Data 2). High reproducibility was observed between technical replicates (130 profiles for 65 samples with adequate material, Supplementary Data 3) across different molecular subtypes, with a median correlation coefficient of 0.9604 (Pearson correlation, ranging from 0.8850 to 0.9748, Supplementary Fig. 1c, d). The nucleosomal periodicity of fragment size, enrichment of ACRs signal at the transcription start site (TSS), and clear signals on representative genes (Supplementary Fig. 1e–g) showed the high quality of the chromatin accessibility profiles generated from primary tumors in this study.

Seventy-five high quality ATAC-seq profiles of 59 patients were obtained after quality control and combine of replicates (Methods). The median number of ACRs identified in each B-ALL sample was 138,366, ranging from 57,941 to 204,563. These ACRs were further combined 758,738 ACRs representing pediatric B-ALL cohort ACRs (c-ACRs, Supplementary Fig. 2a and Supplementary Data 4). We annotated the ACRs to eight functional genomic regions

according to the Epigenomic Roadmap Project[21]. The functional partitioning of B-ALL genome was obtained by analyzing genome-wide histone modifications collected from primary B-ALL cell in Blueprint Epigenomic Consortium with ChromHMM[21] (Methods and Supplementary Fig. 2b). We observed comparable functional distributions for ACRs across B-ALL tumor genomes (Fig. 1b). Genomic regions associated with active gene transcription showed higher chromatin accessibility, in terms of both number and openness of ACRs (Fig. 1b and Supplementary Fig. 2c). ACRs associated with enhancer regions (Enh) accounted for a median of 31.30% of all ACRs in the genome, followed by active transcription site (TssA, 20.38%), transcription-associated regions (Tx, 6.38%), and bivalent Tss/Enh (BivR, 4.24%) (Fig. 1b). On the other hand, transcription repression-related regions showed less accessibility, including PolyComb regions (ReprPC, 8.00%), heterochromatin (Het, 0.87%), and ZNF genes & repeats (ZNF/Rpts, 0.03%) (Fig. 1b). In addition, ACRs in repressive regions were more heterogeneous compared with those in actively transcribed regions (Supplementary Fig. 2d). Surprisingly, Quies regions, which represent genomic regions without well-established histone modifications, also showed chromatin accessibility in B-ALL, accounting for a median of 27.95% of all ACRs (Fig. 1b). This pattern of functional genomic ACRs was also observed at the cohort level when annotating c-ACRs (Supplementary Fig. 2e). Over half (54.94%) of the c-ACRs were found to overlap with Quies regions. To further characterize the Quies regions, we performed H3K27ac ChIP-seq analysis in 12 B-ALL tumor samples (Methods). Results showed that 64.83% of Quies ACRs were located in gene regions (±5% gene length) (Supplementary Fig. 2f). Among these genes, a median of 70.59% also showed H3K27ac signals (Supplementary Fig. 2g) with increased gene transcription (Supplementary Fig. 2h). These data suggested that the ACRs in Quies regions were involved in regulation of active transcription.

### B-ALL-specific chromatin accessible regions associated with leukemogenesis

B-ALL was currently recognized as originating from B-cell precursors[2,22]. We compared the c-ACRs identified above with the chromatin accessibility profiles from previously published pre-pro B and pro B cells, sorted from fetal bone marrow, representing the accessible chromatin status in B-cell progenitors[22]. A down sampling strategy was applied, as the number of ACRs detected was correlated with the sequencing depth in each dataset (Supplementary Fig. 3a). We found that B-ALL showed no significant differences in the quantity of chromatin accessibility across the genome as compared with pre-pro B and pro B cells (Supplementary Fig. 3b). And majority of ACRs detected in pre-pro B cells (98.57%) and pro B cells (98.35%) remained accessible in B-ALL (Fig. 1c). These data supported B-ALL was originated from pre-pro/pro B cells[2,23]. On the other hand, 585,248 (78.39%) ACRs were B-ALL specific. Further analysis found that the B-ALL specific ACRs showed significantly higher heterogeneity compared to the ACRs overlapped between B-ALLs and B-cell progenitors (Fig. 1d, $p < 2.2e{-}16$, Kruskal–Wallis test), consistent with the heterogeneity of chromatin accessibilities in B-ALL tumor cells described above.

We next compared the differential ACRs between B-ALL and B-cell progenitors. A total of 252,028 ACRs showed higher accessibility in B-ALL (Supplementary Data 5). These ACRs were located within the promoter regions (TSS ±1 kb) of 2332 protein coding genes. Enrichment analysis showed that these genes were associated with tumor-related biological processes, including proliferation and differentiation, signal transduction, immune process, cellular response and metabolic process (Supplementary Data 6 and Fig. 1e). Among these genes, there were 61 potential oncogenes including IL7R, TCL1A, TCF3, RHOA and ELL. As showed in Fig. 1f, increased chromatin accessibility was observed in the promoter regions of these oncogenes, indicating a potential regulatory function of these ACRs. Besides, ACRs with

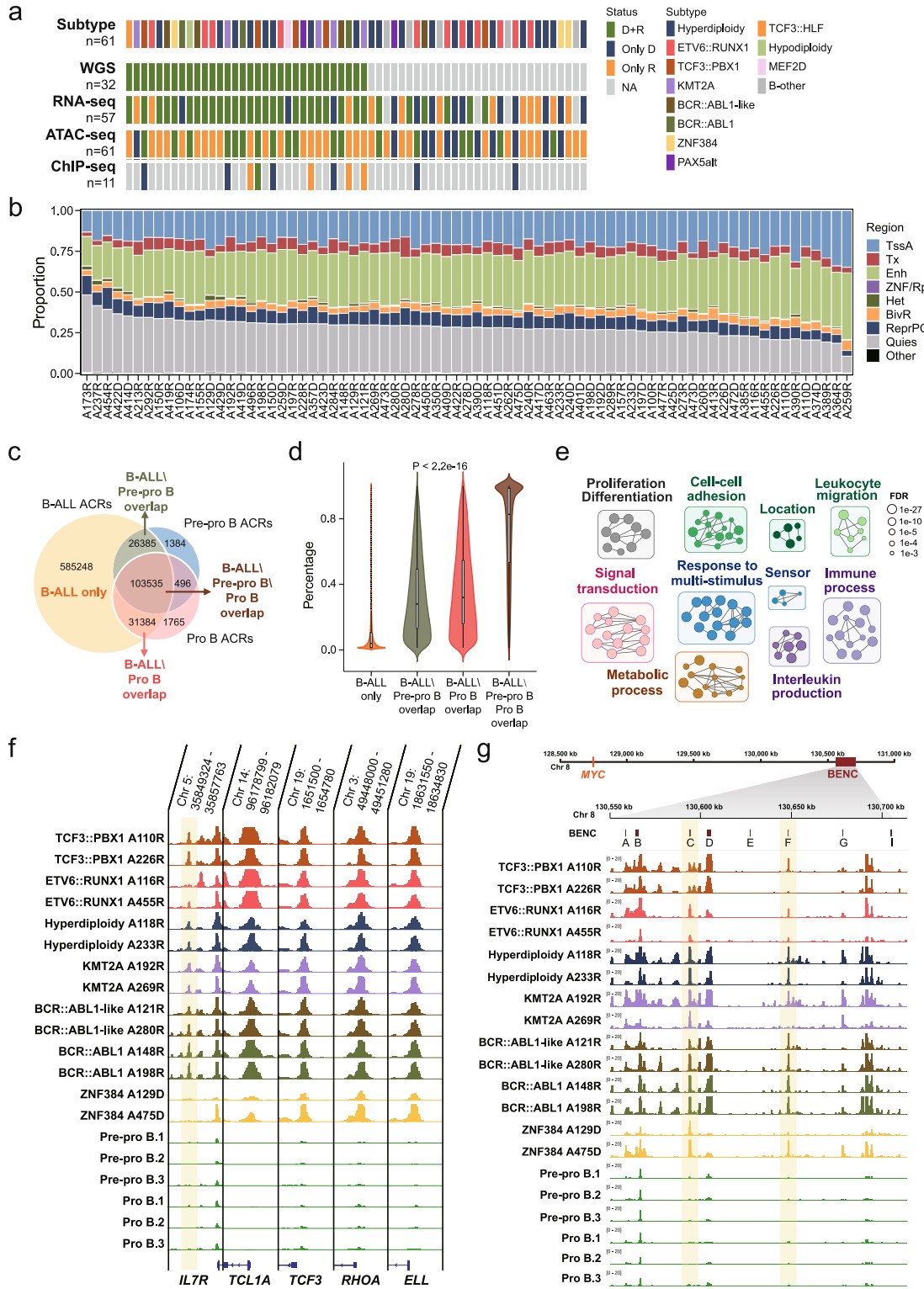

increased chromatin accessibility were also observed in distal enhancer regions. One example presented in Fig. 1g was the distal blood enhancer cluster (BENC), which was reported as a super enhancer that activate *MYC* transcription[24]. Increased chromatin accessibilities were observed in multiple enhancers in this region, consistent with the increased MYC activity in B-ALL[25]. These findings suggested the ACRs with increased accessibility in B-ALL were involved in disease development.

## Chromatin accessibility is associated with subtype-specific transcription regulation in B-ALL

In addition to a difference in ACRs between B-ALL and B-cell progenitors (pre-pro B and pro B cells), we also observed distinguishable differences in chromatin openness among molecular subtypes. As showed in Fig. 2a, B-ALLs were grouped by subtypes when applying unsupervised clustering with recurrent c-ACRs (Methods and Supplementary Data 4). This was supported by calculating pairwise

**Fig. 1 | The patterns of chromatin accessibility in pediatric B-ALL. a** Multi-omics data for 61 patients analyzed in this study. Diagnosis (D) and relapse (R) samples from 11 B-ALL subtypes were analyzed, including hyperdiploidy, ETV6::RUNX1, TCF3::PBX1, KMT2A, BCR::ABL1, BCR::ABL1-like, ZNF384, PAX5alt, TCF3::HLF, hypodiploidy and MEF2D. Cases with unclassified subtype were grouped into B-other. NA not available. **b** The percentages of accessible chromatin regions (ACRs) located in indicated genome regions of B-ALL samples ($n = 75$). **c** Venn diagram shows the overlap between ACRs detected in B-ALL and B-cell progenitors (pre-pro B cells and pro B cells). **d** Violin plot presents the recurrence of ACRs in B-ALLs. The ACRs were grouped into four groups as showed in (**c**). Significant difference was observed among the four groups ($p < 2.2e{-}16$, Kruskal–Wallis test). Box plots show the median number as centers, with upper and lower hinges represent 75th and 25th percentile, and whiskers extend to largest and smallest values no more than 1.5*IQR. (B-ALL only: $n = 585,248$; B-ALL\Pre-pro B overlap: $n = 26,385$; B-ALL\Pro B overlap; B-ALL\Pre-pro B\Pro B overlap: $n = 103,534$). **e** Gene

set enrichment analysis shows that 2332 protein coding genes regulated by 252,028 higher accessible ACRs in B-ALL were enriched in tumor associated biological processes. Only terms with FDR < 0.001 are displayed. The node size represents the enriched FDR values, and the edge represents overlap between two gene sets. Clusters of functionally related categories were manually grouped and labeled in different colors. **f** Wiggle plot shows regions with increased chromatin accessibility in B-ALLs as compared to B-cell progenitors. ACRs with ±1 kb centered the TSS of representative cosmic genes are showed. Only subtypes with more than three cases are included and two samples are randomly selected and showed for each subtype. The ACR present higher accessible in B-ALL upstream TSS of *IL7R* are highlighted in light-yellow. **g** Wiggle plot shows the chromatin accessibility in the blood enhancer cluster (BENC) region (chr8:126,712,193–128,412,193). The positions of enhancers (A–I) are indicated in the BENC track on the top. The tracks showing ACRs in this region are organized as in (**f**). Two enhancers from the BENC cluster showing increased accessibility in B-ALLs are highlighted in light-yellow.

correlations of ACRs between B-ALL samples (Fig. 2b). Subtype-associated accessibility was observed for all functional genomic regions (Fig. 2c and Supplementary Fig. 4); distal regulatory regions including Enh and BivR showed most significant subtype specificity (Fig. 2c), which was consistent with the tissue specificity of distal chromatin open regions reported previously[19]. We next analyzed the difference in ACRs across subtypes. Only subtypes with more than three cases were included in this analysis. Different chromatin accessibilities were observed among subtypes (Supplementary Fig. 5). We found that ETV6::RUNX1 and ZNF384 B-ALL samples showed significantly fewer ACRs compared with other B-ALL samples ($p = 0.0001$, Kruskal–Wallis test). Of the 625,287 recurrent c-ACRs, 17,981 were identified as subtype-specific ACRs (Methods and Supplementary Data 7), with a median of 3083 ACRs in each subtype (range 708–5288). A hierarchical clustering heatmap revealed that these ACRs showed strong subtype-specific accessibility (Fig. 2d).

By combining transcription factor (TF) motif analysis with gene transcription analysis using RNA-seq data (Methods), we identified 109 TFs associated with these subtype-specific ACRs. These TFs were grouped into nine clusters based on their enrichment in each subtype (Fig. 2e). Besides the TFs enriched for a specific B-ALL subtype, we observed high similarity of TF enrichment between some subtypes. This included shared TFs in TCF3::PBX1 and ETV6::RUNX1 subtypes, KMT2A and ZNF384 subtypes, and in BCR::ABL1\BCR::ABL1-like and hyperdiploidy subtypes. Some of these observations were supported by previous reports. For example, overlapped between the KMT2A rearranged and ZNF384 B-ALL subtypes is concordant with the fact that both subtypes show a tendency of myeloid transcription[26–28]. While the mechanisms remained further investigated, the shared transcription regulation suggested a similarity of cell differentiation states between subtypes.

To further explore the regulatory role of these TFs, we performed expression analysis on both the TF and their potential target genes between tumor samples of the enriched subgroup versus the others. Fourteen TFs were found with significantly increased transcription in the enriched subtype ($p < 0.05$ and FC > 1.2) (Fig. 2f), suggesting the transcription regulation directly associated with the TF expression. For the target gene analysis, we focused on the 53 TFs with the binding motif in gene promoter regions (TSS ± 1 kb), and these genes were analyzed as the targets for each transcription factor (Methods). The results of individual target gene analysis were further combined to represent the regulatory function of the transcription factor in the enriched subtype. As showed in Fig. 2g, 13 out of the 53 transcription factors included in this analysis were found with significantly higher expression of the target genes ($p < 0.05$ and FC > 1.2), supporting the increased transcription regulation activity in the enriched subtype. We noticed that 12 out of 13 TFs in the target gene analysis did not show expression changes of the TFs themselves, indicating a context dependent transcription regulation among B-ALL subtypes. Among 13

TFs, *E2F6* was identified as specifically enriched in the ETV6::RUNX1 subtype. This gene plays a crucial role in the control of the cell cycle and is associated with tumor growth or chemotherapy sensitivity in a variety of tumors[29–31]. Concordantly, significantly higher transcription of both *E2F6* and its target genes was observed in the ETV6::RUNX1 B-ALL subtype compared with other subtypes (Fig. 2f, g). These results provided further insights into the subtype specific transcription regulation in B-ALL.

## Allele-specific open chromatin in B-ALL is associated with leukemia

A total of 44 samples (including 13 diagnosis samples and 31 relapse samples) from 32 B-ALL patients with paired ATAC-seq and WGS data were analyzed for allele-specific open chromatin (ASOC). A median of 3616 ASOC regions were identified per sample (Supplementary Data 8A). ASOC regions accounted for a median of 14.39% of ACRs genome wide, which was significantly less than the biallelic open chromatin (BiOC) regions (median 85.61%, $p < 2.2e{-}16$, Wilcoxon test, Fig. 3a). Moreover, fewer ASOC regions were found in active transcription-related regions (TssA, Tx, and Enh) compared with BiOCs ($p < 2.2e{-}16$, Fisher's exact test, Fig. 3b). Further analysis showed that ASOC ACRs tended to be closer together compared with BiOC ACRs (Supplementary Fig. 6a) and more likely to be grouped into a single topological associated domain (TAD, Supplementary Fig. 6b). These data suggested that the regulation of chromatin accessibility between alleles fitted into the regulation of the three-dimensional genome architecture.

We next investigated chromatin accessibility using leukemia-associated single nucleotide polymorphisms (SNPs) from EpiMap[32]. A total of 46 leukemia-related SNPs with imbalanced chromatin accessibility between alleles were found in ASOC regions (Supplementary Data 8B), including 7 SNPs present in at least 5 samples (Fig. 3c, d). Among these top recurrent SNPs was rs7090445, which was previously predicted to reduce the transcription of *ARID5B* by disrupting RUNX3 binding with the C-allele[33]. Interestingly, we observed that the chromatin accessibility of the T-allele of this locus was significantly higher than that of the C-allele in 14 out of 21 (66.67%) of B-ALL samples with heterozygous C/T alleles, consistent with the role of the C-allele in leukemia. Another recurrent SNP was rs13401811, with G-allele previously reported as the risk allele in chronic lymphocytic leukemia. ATAC-seq data showed that the G-allele had significantly higher chromatin accessibility than the A-allele in 8 out of 9 (88.89%) B-ALL samples with the G/A genotype. It is noteworthy that rs13401811 is located -262 kb upstream of *BCL2L11*, which encodes a pro-apoptotic protein that is involved in ALL drug resistance[20]. These results indicated that chromatin accessibility is associated with the function of these disease-associated SNPs.

In addition, we identified 556 COSMIC genes in the neighborhood of these ASOC regions (Methods, Supplementary Data 8C). Notably,

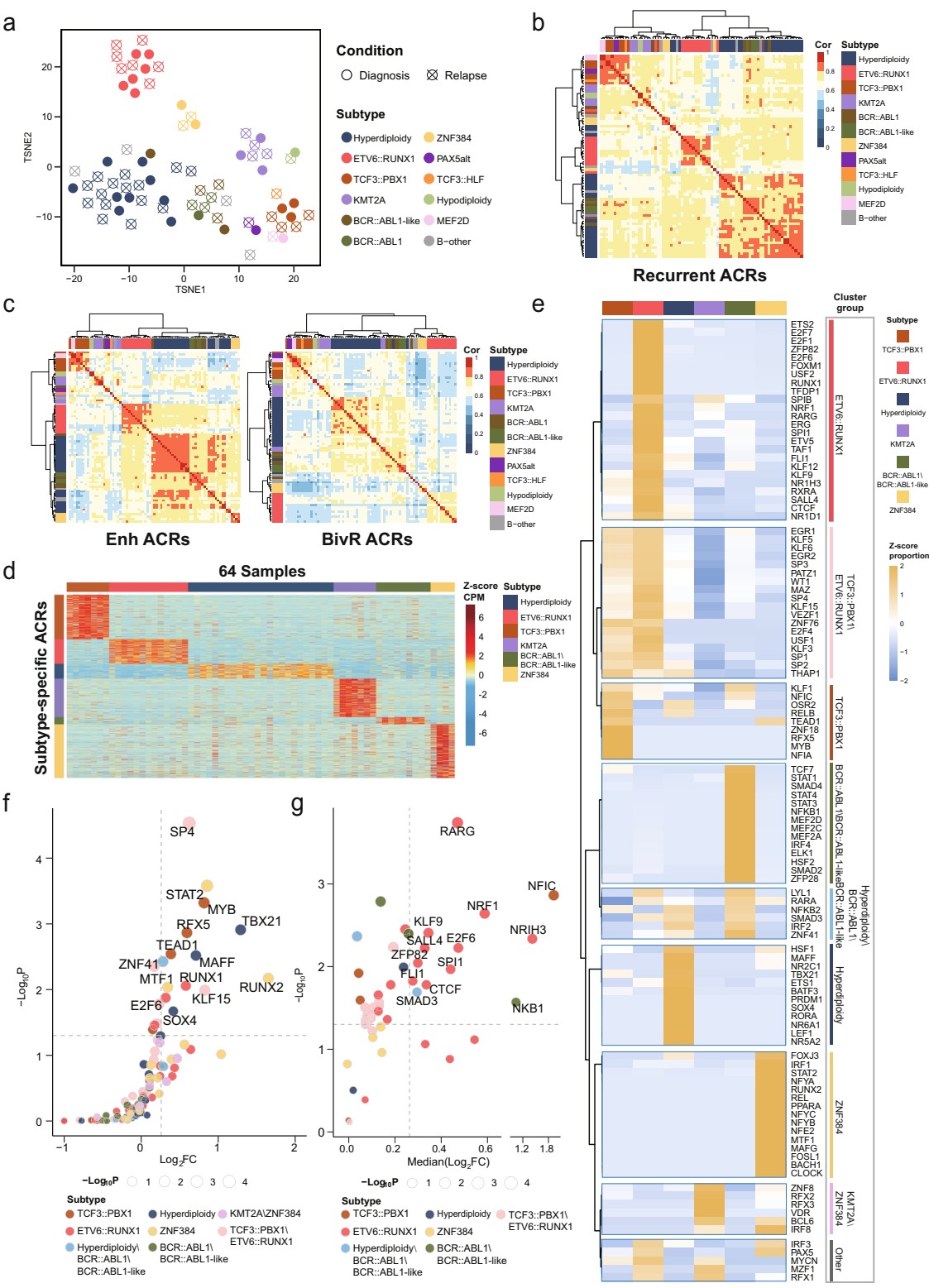

allele-specific transcription was simultaneously detected in some of these potential oncogenes, such as *MECOM* and *HOXA9* (Supplementary Fig. 6c). Broad imbalanced chromatin accessibility between alleles was observed for both genes, suggesting ASOC was involved in the cis-activation of these genes.

**Chromatin accessibility changes in response to B-ALL treatment**
We analyzed the ACRs with differential accessibility ($|\log_2 FC| > 1$ and false discovery rate [FDR] <0.05) between diagnosis and relapse for

each subtype (Supplementary Data 9). Only subtypes with more than five diagnosis or relapse samples were included in this analysis. A median of 945 differential ACRs in each subtype were identified, ranging from 268 to 4072 (Fig. 4a). Notably, only 1.54% (91 out of 5911) of ACRs with higher accessibility in the relapse samples (relapse-high) and 0.14% (2 out of 1423) with lower accessibility in the relapse samples (relapse-low) were shared between two or more subtypes, indicating significant heterogeneity in chromatin accessibility changes during treatment among different subtypes (Fig. 4b).

**Fig. 2 | Subtype-specific chromatin accessibility in B-ALL. a** The t-distributed stochastic neighbor embedding (t-SNE) plot showing the clustering of 75 B-ALL samples based on recurrent c-ACRs with top 10% highest variance. **b** Heatmap of Pearson correlation coefficients shows the inter-sample correlation of chromatin accessibility based on all recurrent ACRs. **c** Heatmaps of Pearson correlation coefficients based on ACRs in Enh regions (left) and ACRs in BivR regions (right) showing the inter-sample similarity of chromatin accessibility. **d** The accessibility of 17,981 subtype-specific ACRs in 64 B-ALL samples is shown in heatmap. The *x* axis presents 64 B-ALL samples and *y* axis displays subtype-specific ACRs. **e** Unsupervised clustering of 109 transcription factors (TF) based on their enrichment in each subtype, as labeled on the right of this plot. **f** Differential expression analysis of subtype enriched TFs in B-ALL. The TFs are colored according to the clusters in (**e**). For each TF, statistical analysis was performed to test the expression

difference between B-ALLs of the enriched subgroup versus the others. Each dot represents an individual TF. Horizontal dashed line represents $p = 0.05$ and vertical dashed line represents fold change (FC) = 1.2. Gene symbols of TFs with $p < 0.05$ (one-sided Wilcoxon test) and FC > 1.2 are showed. **g** Expression analysis of target genes of subtype enriched TFs. Only TFs with binding motif in gene promoter region (TSS ± 1 kb) were included. Differential expression analysis was performed for each target gene between B-ALLs grouped upon the enriched subgroups of TF. The median fold change (FC) of all target genes regulated by the individual TF is showed on *x* axes. Each dot represents a group of target genes for an individual TF. Horizontal dashed line represents $p = 0.05$ and vertical dashed line represents fold change (FC) = 1.2. Gene symbols of TF with target genes satisfied geometric mean $p < 0.05$ (one-sided Wilcoxon test for each target gene) and FC > 1.2 are labeled.

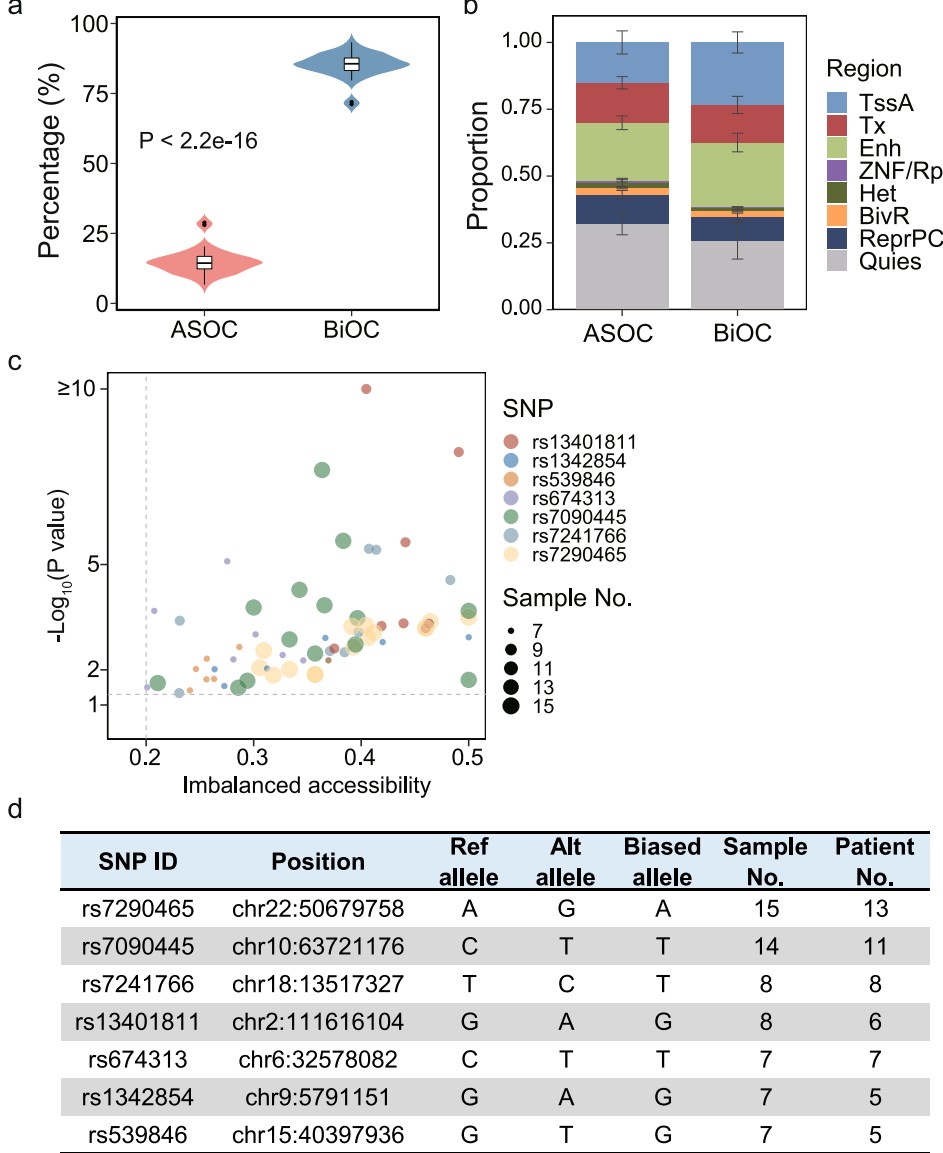

| SNP ID | Position | Ref allele | Alt allele | Biased allele | Sample No. | Patient No. |
|---|---|---|---|---|---|---|
| rs7290465 | chr22:50679758 | A | G | A | 15 | 13 |
| rs7090445 | chr10:63721176 | C | T | T | 14 | 11 |
| rs7241766 | chr18:13517327 | T | C | T | 8 | 8 |
| rs13401811 | chr2:111616104 | G | A | G | 8 | 6 |
| rs674313 | chr6:32578082 | C | T | T | 7 | 7 |
| rs1342854 | chr9:5791151 | G | A | G | 7 | 5 |
| rs539846 | chr15:40397936 | G | T | G | 7 | 5 |

**Fig. 3 | The allele-specific open chromatin (ASOC) regions in B-ALL. a** Violin plot showing the percentage of ASOC regions and biallelic open chromatin (BiOC) regions in each B-ALL sample. (32 samples, one sample per patient, $p < 2.2e{-}16$, two-sided Wilcoxon test). Box plots show the median number as centers, the upper and lower hinges represent 75th and 25th percentile, and whiskers extend to largest and smallest values no more than 1.5*IQR. **b** Bar plot showing the proportion of ASOC and BiOC regions that are found in different genomic regions. Data are presented as mean values ± SD. (32 samples, one sample per patient). **c** Scatter plot presenting the recurrent ASOCs linked to leukemia-associated SNPs. Each dot

represents one ASOC-SNP pair in an individual B-ALL sample, and the size of the dot represents the number of samples carrying the ASOC-SNP. The significance of different chromatin accessibility between two alleles is shown on the *y* axes as −log₁₀(*p* value) (Methods) with dashed line represent a *p* value of 0.05. The difference of chromatin accessibility between two alleles is shown as absolute value on *x* axes, with dashed line represent a difference of 0.2 (Methods). Different SNPs are represented by different colors. (44 samples of 32 patients). **d** The table displays detailed information about the seven SNPs shown in (**c**).

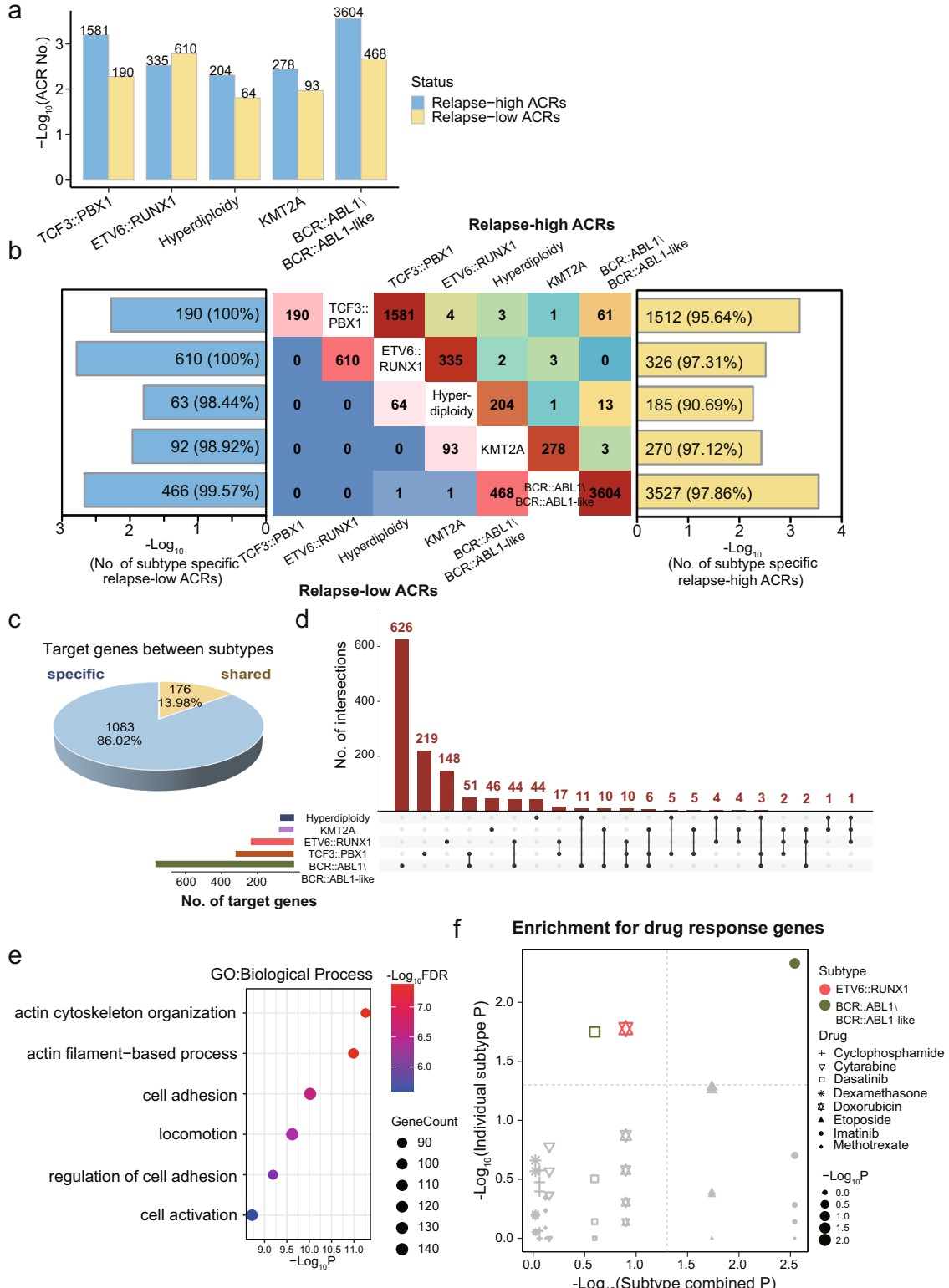

**Fig. 4 | Chromatin accessibility alterations in relapsed B-ALL patients. a** The number of differential ACRs between tumors from relapsed and diagnosed B-ALL patients ($|\log_2 FC| > 1$ and FDR < 0.05). Relapse-high ACRs and Relapse-low ACRs represent ACRs with increased and decreased accessibility in relapse tumor cells, respectively. **b** Heterogeneity of diagnosis-relapse differential ACRs among B-ALL subtypes. The heatmap in the middle shows the number of ACRs shared between subtypes, with the number of relapse-high ACRs shown on the upper right and number of relapse-low ACRs on the bottom left. The bar plots present the percentage of subtype-specific relapse-low ACRs (left) and relapse-high ACRs (right) in each subtype. **c** Pie chart showing the percentage of shared and subtype-specific target genes regulated by diagnosis-relapse differential ACRs predicted by ACR-to-gene links. **d** Upset plot showing the target genes shared across subtypes. **e** Enrichment analysis shows that target genes of differential ACRs are enriched for cell adhesion-related biological process terms. The top six terms with the most significant FDR are listed. **f** Target genes of diagnosis-relapse differential ACRs are enriched in drug-response genes. Significant enrichment ($p < 0.05$) was determined by two-sided Fisher exact test. The horizontal axis shows the enrichment analysis results for target genes associated with diagnosis-relapse differential ACRs for all subtypes combined. The vertical axis displays the enrichment analysis results for individual subtypes.

To obtain the target genes of these differential ACRs that are dysregulated during relapse, we performed ACR-to-gene predictions on a total of 52 B-ALL samples with paired ATAC-seq and RNA-seq data (Methods) and defined 116,307 ACR-gene correlations (Supplementary Data 10). With this, a total of 1259 genes were identified as being potentially targeted by the ACRs dysregulated during relapse (Supplementary Data 11). As expected, significant heterogeneity was observed, with only 13.98% (176 out of 1259) of target genes shared between any two subtypes (Fig. 4c, d). Enrichment analysis suggested that these target genes were associated with cell adhesion-related biological processes (Fig. 4e), suggesting potential dysregulated interaction of leukemia cells and mesenchymal stromal cells in the bone marrow microenvironment, which was previously shown to be associated with chemoresistance of leukemia cells[34,35].

On this basis, we integrated the drug susceptibility data from CTD^2[36] to investigate whether the observed dysregulation of genes in relapsed B-ALL patients was associated with clinical treatment. Firstly, we determined the association between gene transcription data from 11 B-ALL cell lines (collected from the CCLE project) with the cell response to 8 drugs commonly used in B-ALL treatment, including Cytarabine and Methotrexate. This analysis resulted in a total of 14,680 drug-gene pairs representing the transcriptional alterations associated with drug response (Methods and Supplementary Data 12). Interestingly, the potential target genes regulated by the relapse associated ACRs were significantly correlated with drug treatments, including Imatinib ($p = 0.0029$, Fisher's exact test) and Etoposide ($p = 0.0184$, Fisher's exact test) (Fig. 4f). Similar results were observed when we performed the analysis for drug-gene pairs identified within individual B-ALL subtypes. Target genes of differential ACRs of BCR::ABL1\B-CR::ABL1-like subtype were significantly correlated with Imatinib ($p = 0.0047$, Fisher's exact test) and Dasatinib ($p = 0.0178$, Fisher's exact test), both of which are tyrosine kinase inhibitors used in BCR::ABL1\BCR::ABL1-like B-ALL treatment, whereas a significant association with Doxorubicin was observed for the ETV6::RUNX1 subtype ($p = 0.0167$, Fisher's exact test) (Fig. 4f). These results indicated that the treatment could reshape chromatin accessibility to impact gene transcription regulation during B-ALL relapse.

## Chromatin accessibility features affect patient outcomes

We analyzed relapse-free survival (RFS) to investigate how chromatin accessibility correlates with B-ALL prognosis. ATAC-seq data for 42 patients with relapsed B-ALL treated with CCCG-ALL-2009 ($n = 37$) and CCCG-ALL-2015 ($n = 5$) protocols were analyzed. No significant difference of patients' prognosis was observed between the two protocols (Supplementary Fig. 7a). A total of 70,573 (11.29%) RFS-related ACRs (FDR < 0.05) out of 625,287 recurrent c-ACRs were identified (Supplementary Data 13). Potential targets of these RFS-related ACRs as predicted from ACR-to-gene links were enriched in cell cycle and leukocyte differentiation-associated biological processes (Fig. 5a), suggesting the regulation of these ACRs on the proliferation and differentiation of B-ALL blasts. Unsupervised clustering of the 42 relapse B-ALL patients using the RFS-related ACRs resulted in two B-ALL groups (Group A and Group B) with distinct times to relapse and prognoses (Fig. 5b, c). Concordant results were observed with different clustering methods (Supplementary Fig. 7b). Interestingly, a similar pattern was observed for the matched diagnosis samples (Supplementary Fig. 7c). To validate this observation, we analyzed data from the Therapeutically Applicable Research to Generate Effective Treatments (TARGET) project[37]. We focused on the RFS-associated ACRs with higher accessibility in B-ALLs from Group B ($n = 10,975$, $|log_2FC| > 1$ and FDR < 0.05) and found 1827 potential target genes from ACR-to-gene association analysis (Supplementary Data 14). With these genes, 252 B-ALL samples from TARGET project with prognosis information (48 patients were diagnosis-relapse paired) were analyzed and grouped into 3 clusters (Supplementary Fig. 7d and Supplementary Data 15). Survival analysis

showed significant differences in both event-free survival (EFS) and overall survival (OS) between the clusters (Supplementary Fig. 7e). Patients showed the highest expression of the target genes showed the worst prognosis (Cluster 3). This result served as independent validation that aberrations in chromatin accessibility reflected patient prognosis.

Consistent with a previous report[38], Group A patients were enriched in the ETV6::RUNX1 and hyperdiploidy subtypes and showed relatively good prognoses, while Group B patients were mostly KMT2A and BCR::ABL1\BCR:ABL1-like subtypes and showed inferior prognoses (Fig. 5b, c). Notably, although TCF3::PBX1 B-ALL patients are generally considered low risk[39], all four TCF3::PBX1 cases in this analysis were grouped with the KMT2A and BCR::ABL1\BCR::ABL1-like cases in Group B and relapsed within 22 months from diagnosis (Fig. 5b). Concordantly, 19 out of 21 TCF3::PBX1 B-ALLs of TARGET project were grouped into Cluster3 with worst prognosis in above-mentioned analysis (Supplementary Data 15). Interestingly, hyperdiploidy B-ALL cases were separated into two different groups (Fig. 5b). Three relapsed hyperdiploidy cases (A118R, A174R, and A233R) were clustered in Group B with KMT2A and BCR-ABL1\BCR::ABL1-like and showed inferior RFS (Fig. 5b, c). A total of 7566 differential ACRs were identified between the two hyperdiploidy subgroups ($|log_2FC| > 1$ and FDR < 0.05) (Fig. 5d and Supplementary Data 16). Among these 3156 ACRs showed increased chromatin accessibility in hyperdiploidy cases with inferior RFS in Group B. We obtained 603 potential target genes of these ACRs from the results of ACR-to-gene links. Further analysis showed that these target genes mimic an expression feature of stem cells and myeloid progenitors, including megakaryocyte-erythrocyte progenitors and granulocyte-macrophage progenitors (Fig. 5e). Functionally, the target genes were found to be enriched in migration/adhesion/locomotion-related categories, which are associated with drug resistance[34]. This indicated that the three hyperdiploidy B-ALL patients showed increased lineage plasticity with more myeloid-like and inferior treatment responses compared with other hyperdiploidy B-ALL patients. We further analyzed this ACR pattern associated with inferior prognosis of hyperdiploidy B-ALLs from TARGET cohort. Forty-three hyperdiploidy samples (7 patients were diagnosis-relapse paired) were clustered into 3 clusters based on the 603 target genes as described above (Fig. 5f). Cluster 3, which consisted of five samples collected from three patients, showed the highest expression of the target genes representing the high-risk group. Accordingly, patients of Cluster 3 had the worst prognosis in terms of both EFS and OS (Fig. 5g), supporting our observations. However, the number of patients in this analysis was limited and the aberrant ACRs remained further investigation with a large cohort.

## Discussion

Although the regulatory elements in human genome are well-recognized to play important roles in gene transcription regulation[15,40], the understanding of their function and aberrations in diseases are lagging genome sequencing analysis. Transcription dysregulation is one of the key aberrations in pediatric leukemia given that ancestral fusions in tumor cells usually involve core TF genes in hematopoiesis[41,42]. Analysis of chromatin accessibility in primary tumors using ATAC-seq has provided important information linking transcription dysregulation to the genome and expanding the understanding of genomic and epigenomic evolution in cancer[19,43]. In the present study, we depicted the landscape of chromatin accessibility in 61 B-ALL patients using ATAC-seq. By comparing to B-cell progenitors, we identified a group of ACRs with increased accessibility in B-ALL with target genes enriched for tumor associated processes (Fig. 1e), supporting the hypothesis that chromatin accessibility involved in transcription dysregulation and plays an important role in this disease.

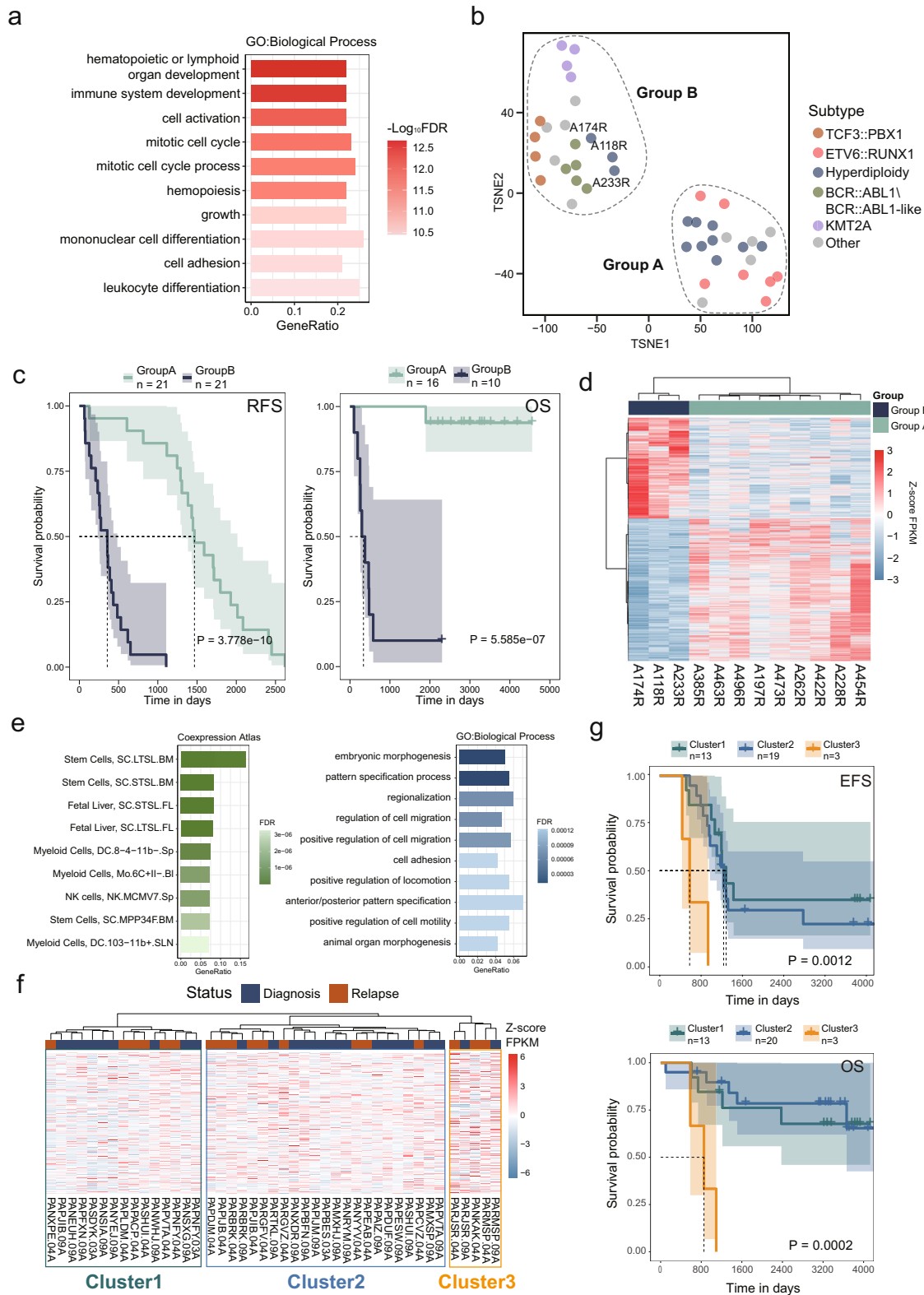

We constructed the functional partitioning of genome by analyzing the public available histone modification data from primary B-ALL cells (Supplementary Fig. 2b) and use this information to annotate the accessible chromatin regions in B-ALL. Surprisingly, a median of 27.95% of ACRs in each individual B-ALL sample were in Quies regions (Fig. 1b) which were without well-established histone modifications to date[21]. Since the functional partitioning of genome used as reference in this analysis was constructed from only one B-ALL

patient, this observation raised the possibility that histone modifications were acquired de novo in these regions in individual B-ALL. This was verified by using H3K27ac ChIP-seq data profiling active enhancers. We showed that H3K27ac modifications indeed overlapped with Quies ACRs, suggesting the potential regulatory function of these Quies ACRs in B-ALL. These observations provided evidence of chromatin accessibility rewiring in tumorigenesis. However, many Quies ACRs do not overlap with actively histone modification. The function

**Fig. 5 | Chromatin accessibility features associated with B-ALL prognosis.**
**a** Gene set enrichment analysis shows that 3976 target genes regulated by 70,573 RFS-related ACRs are enriched for hematopoietic development and cell cycle-associated biological process terms. The categories with the most significant FDR values are listed and are sorted by FDR values in reverse order. **b** T-SNE plot showing the unsupervised clustering of 42 relapse B-ALL samples based on the top 10% most significant RFS-related ACRs. B-ALL samples were clustered into Group A and Group B. **c** The relapse-free survival rate (RFS, $n = 42$) and overall survival rate (OS, $n = 26$) estimates for B-ALL samples of Group A and Group B. (Log-rank test, $p = 3.778e-10$ for RFS and $p = 5.585e-7$ for OS, the error bands indicate 95% confidence intervals, and the dash line indicate the median survival time). **d** Unsupervised clustering based on the CPM of differential ACRs between three hyperdiploidy cases in Group B and nine hyperdiploidy cases in Group A. **e** Gene set enrichment analysis indicates that 603 target genes regulated by 3156 ACRs up-regulated in the hyperdiploidy cases in Group B were enriched for stem cell- and myeloid progenitor-associated gene expression signatures (left panel) and cell adhesion/migration-associated biological process terms (right panel). The top categories with the most significant FDR values are listed. **f** Unsupervised clustering analysis of 43 hyperdiploidy cases from the TARGET project based on the expression of 603 target genes regulated by 3156 ACRs up-regulated in Group B. **g** Event-free survival rate (EFS, $n = 35$) and overall survival rate (OS, $n = 36$) estimates for hyperdiploidy B-ALL samples from the TARGET project. Cases in Cluster3 with high expression of the target genes and mimicking the hyperdiploidy B-ALL cases in Group B had worse outcomes. (Log-rank test, EFS $p = 0.0012$ and OS $p = 0.0002$, the error bands indicate 95% confidence intervals, and the dash line indicate the median survival time).

of these regions in B-ALL remains unknown. Recently, novel histone modifications have been identified, including histone lactylation[44] and serotonylation[7] among others. Further investigation of Quies ACRs for the presence of these emerging histone modifications might provide more insights into the rewiring of the transcriptional regulatory landscape in cancer.

Previous studies have identified somatic mutations in a group of 12 genes that are involved in drug-resistant relapse of leukemia[5]. However, only 13 out of 32 patients with WGS data in current study (40.63%) were found to carry these mutations, leaving the molecular cause unclear for over half of the relapsed B-ALL patients. We investigated the genome-wide chromatin accessibility of 29 diagnosis and 46 relapse B-ALL cases to tackle this question. Genome-widely, we did not observe a large proportion of chromatin accessibility changes between diagnosis and relapse tumor cells, with only 5911 relapse-high ACRs (0.95% of B-ALL recurrent ACRs) and 1423 relapse-low ACRs (0.23%) identified. The small number of ACR changed might partially be due to the high heterogeneity of chromatin accessibility among B-ALL subtypes, which we showed in this study. Meanwhile, drug treatment signatures were observed within these differential ACRs. The genes potentially targeted by these ACRs were significantly associated to genes involved in the response to drugs commonly used in B-ALL treatment, including Doxorubicin and Etoposide (Fig. 4f). Interestingly, our data showed an association between chromatin accessibility changes and targeted therapy with tyrosine kinase inhibitors. Significant associations between ACR changes and Dasatinib/Imatinib were observed particularly in BCR::ABL1\BCR::ABL1-like B-ALL samples, in line with the fact that these drugs are widely used for treating BCR::ABL1\BCR::ABL1-like B-ALL patients in clinic[45]. In addition, enrichment of genes involved in the response to Doxorubicin was only observed for the ETV6::RUNX1 subtype, indicating subtype-specific response. These data suggest that drug treatment might reshape the chromatin accessibility landscape of tumor cells. As clonal evolution is common during leukemia treatment, experiment simultaneously analyze the chromatin accessibility and gene mutations at single cell level for paired diagnosis and relapsed tumors would provide further information regarding the association between chromatin accessibility changes during treatment and clonal evolution.

Survival analysis discovered over 70,000 ACRs significantly associated with RFS. A particularly notable finding was that using RFS-associated ACRs, B-ALL patients could be clustered into two groups with distinct prognoses, indicating the effect of chromatin accessibility regulation on tumor progression (Fig. 5b, c and Supplementary Fig. 7b). Surprisingly, B-ALL patients of hyperdiploidy subtype, which is usually associated with good prognosis[46], were split into two groups and showed distinct prognoses (Fig. 5b, c). This observation was further validated independently in the TARGET B-ALL cohort by analyzing the potential target genes dysregulated by these RFS-associated ACRs (Fig. 5f, g). As hyperdiploidy B-ALL accounts for over 30% of B-ALL cases, these findings might lead to the identification of a number of high-risk B-ALL patients in a relatively low-risk subgroup. Precise risk

classification taking this chromatin accessibility pattern into account would ensure that patients receive proper treatment and further improve the prognosis of B-ALL patients.

The genes potentially targeted by the differential ACRs with increased accessibility in the hyperdiploidy B-ALL patients with inferior prognoses showed stem cell and myeloid progenitor-like signatures. These data indicated an increased potential of lineage plasticity for hyperdiploidy B-ALL patients in this group. Lineage plasticity in cancer refers to the lineage transition of cancer cells under selective pressure such as clinical treatment. This phenomenon has been described in several recent studies and is associated with drug resistance[47,48], including prostate cancer[49] and lung cancer[50] among others. Our data here suggested that lineage plasticity also exists in B-ALL and is associated with treatment resistance. The leukemia cells underwent a transition toward being more stem-cell like under the pressure of treatment and resulted with alternated extracellular bone marrow microenvironment dependencies and intracellular transcription regulatory circuit as showed in Fig. 5e, leading to treatment resistance and relapse. Importantly, this transition was shared between diagnosis and relapse tumor cells, providing the opportunity to predict early relapse by analyzing diagnosis samples and develop alternative therapeutic strategies accordingly.

We showed that there is high heterogeneity in chromatin accessibility in B-ALL patients, which requires the investigation of more cases in the future. In addition, all the cases investigated here were relapsed cases resulting in the chromatin accessibility profiles being biased toward high-risk B-ALL. Recently, there was one pre-published study profiling the chromatin accessibility of B-ALL[51]. A more comprehensive analysis that combines these data and includes more standard risk B-ALL patients would provide further evidence of the chromatin accessibility aberration in B-ALL. Nevertheless, we have presented the landscape of chromatin accessibility in pediatric B-ALL and characterized ACRs specifically enriched in B-ALL patients and in different molecular subtypes. More importantly, we showed the occurrence of chromatin remodeling under drug treatments and identified the chromatin accessibility signatures associated with early relapse. These results expand our understanding of genomic aberrations behind B-ALL and highlight the importance of epigenomic features for risk stratification of this malignancy.

## Methods
### Patient samples
Bone marrow samples were obtained from 61 relapsed B-ALL patients treated through 2007–2019 in Shanghai Children's Medical Center (SCMC). Patients were treated under ALL-SCMC-2005 protocol ($n = 4$), ALL-SCMC-2009 protocol ($n = 46$) and ALL-SCMC-2015 protocol ($n = 11$, Supplementary Data 1). Among the patients, 17 were diagnosed under the age of 3 years, 30 were between the ages of 3–10 years, and the remaining 14 patients were 10–15 years. This study was approved by the Shanghai Children's Medical Center Institutional Review Board. Informed written consents were obtained from parents for all patients.

## Subtype classification

The molecular subtypes of B-ALLs were classified by combining the results of following analysis: (1) gene expression pattern-based subtype classification by our in-house developed recurrent neural network (RNN) based model (Cui B., Sun H., Wang H., Zhao S., Rao J., Wu W., Wang R., Fan R., Li B., Shen S., Liu Y., manuscript in preparation), (2) fusions, structure variations and driver mutations detected in RNA-seq and/or whole genome sequencing data, (3) the CNV results from whole genome sequencing and RNA-seq analysis, (4) karyotyping from clinical test. For each individual case, results from all above-mentioned analyses were collected and manually curated for subtype classification. The resulted molecular subtype will be cross validated for the cases with both diagnosis and relapsed samples analyzed.

## Enrichment of high viability leukemia cells with FACS

The cryopreserved leukemia cells were thawed in 37 °C water bath and transferred to RPMI 1640 culture medium. Cell clumps after centrifugation were treated with DNaseI (Sigma, DN25) to digest the DNA released by dead cells. The LIVE/DEAD™ Fixable Dead Cell Stain Kits (Invitrogen, L23101) was used to distinguish living cells, and lineage-associated antibodies anti-human CD19 conjugated with APC (Bioscience,17-0199-42), anti-human CD10 conjugated with PECY7 (Biolegend, 312214) and anti-human CD45 conjugated with APC-CY7 (Biolegend, 304014) were selected to enrich tumor cells according to the immunophenotyping reports of each patient at diagnosis (Supplementary Data 2). After staining for 30 min in dark at 4 °C, living leukemia cells were sorted by FACS (Beckman, MoFlo XDP) for downstream experiments.

## ATAC-seq

ATAC-seq was performed according to the methods as previously reported[18]. To prepare nuclei, we washed 50,000 sorted cells with cold 1x PBS and centrifugation at $500 \times g$ for 5 min. The cells were resuspended with 50 µl cold ATAC-resuspension buffer (RSB) (10 mM Tris-HCl PH 7.4, 10 mM NaCl, 3 mM MgCl2 in nuclease free water) containing 0.1% NP40, 0.1% Tween-20 and 0.01% Digitonin, followed by lysis on ice for 10 min. After lysis, we added 1 ml RSB containing 0.1% Tween-20, spun nuclei at $500 \times g$ for 10 min. Immediately following the nuclei prep, the nuclei pellet was resuspended in the transposase reaction mix [10 µl TruePrep Tagment Buffer L (Vazyme, TD501), 5 µl TruePrep Tagment Enzyme (Vazyme, TD501), 16.5 µl PBS, 0.5 µl 1% digitonin, 0.5 µl 10% Tween-20 and 17.5 µl nuclease free water]. The transposition was incubated at 37 °C for 30 min in a thermomixer with 1000 RPM mixing. DNA from transposition reaction was purified with DNA Clean and Concentrator-5 Kit (Zymo, D4014) and eluted in 21 µl elution buffer. The eluted DNA was amplified with TruePrep DNA Library Prep Kit V2 for Illumina (Vazyme, TD501) and TruePrep Index Kit V2 for Illumina (Vazyme, TD202). SPRI size selection was performed with VAHTS DNA Clean Beads (Vazyme, N411) to exclude fragments larger than 1200 bp. All libraries were sequenced using paired-end, dual-index sequencing on Illumina NovaSeq 6000.

## RNA-seq

Total RNA was extracted from fresh frozen tumor cells with TRIzol. RNA integrity was assessed using Agilent Bioanalyzer 2100 system and RIN value (>6) was request for library construction. Ribo-Zero strand-specific library was adopted for samples with a total mass greater than 2 µg, and mRNA-seq library was adopted for other samples (Supplementary Data 1). For strand-specific library construction, ribosome RNA was removed from total RNA by NEBNext rRNA Depletion Kit (NEB, #E6310). For mRNA-seq library, poly-A mRNA was purified from total RNA using NEBNext Poly(A) mRNA Magnetic Isolation Module (NEB #E7490). Sequencing libraries were generated using NEBNext® UltraTM RNA Library Prep Kit for Illumina (NEB, #E7530) following manufacturer's recommendations and index codes were added to

attribute sequences to each sample. The purified cDNA libraries were sequenced on the Illumina NovaSeq 6000 system with PE-150 bp.

## ChIP-seq

In total, $3 \times 10^6$ leukemia cells sorted by FACs in 400 µl PBS were fixed with 1% formaldehyde (CST, 12606) at room temperature for 10 min on a rotator, and 0.125 M Glycine was added to stop the cross-linking reaction for 5 min. The cells were resuspended in cold lysis buffer (10 mM Tris-HCl pH 7.5, 10 mM NaCl, 3 mM MgCl₂ and 0.5% NP-40) after washing and rotated 10 min at 4 °C. Chromatin pellets obtained by centrifugation at $1700 \times g$ for 5 min were washed twice with 300 µl sonication buffer [10 mM Tris-HCl pH 8.0, 1 mM EDTA pH 8.0, 0.1% SDS, 3 µl Protease/Phosphatase Inhibitor Cocktail (CST, 5872)] and resuspended with 120 µl sonication buffer in microTUBE, followed by sonication with Covaris M220 sonicator for 15 min at 7 °C until the size of most fragments was in the range of 200–700 bp. Sonicated chromatin was rotated at 4 °C for 2 h with 5 µl of anti-histone H3K27ac antibody (Abcam, 4729), 2 µl spike-in antibody (Active motif, 61686) and 5 µl spike-in chromatin (Active motif, 53083). Dynabeads Protein G (Life Technologies, 10003D) was added followed by incubation at 4 °C overnight on a rotator. Beads were washed twice with cold RIPA buffer (50 mM Tris-Cl PH = 7.5, 300 mM NaCl, 1.0% Triton X-100, 0.5% sodium deoxycholate and 0.1% SDS) and additional three times with cold LiCl washing buffer (100 mM Tris-HCl pH 7.5, 500 mM LiCl, 1% NP-40 and 1% sodium Deoxycholate). Chromatin precipitated was then incubated with elution buffer (50 mM Tris-Cl PH 7.5, 10 mM EDTA, 0.1% SDS, 200 mM NaCl) containing 2 mg/ml Proteinase K at 65 °C overnight, to revert formaldehyde cross-linking. Finally, the ChIPed DNA fragments were purified using a DNA Clean and Concentrator-5 Kit (Zymo, D4014) and sent for high-throughput sequencing at Novogene.

## ATAC-seq data processing

ATAC-seq data analysis was performed as previous described[18]. In brief, FastQ Screen[52] (v0.13.0) and FastQC (v0.11.9, https://www.bioinformatics.babraham.ac.uk/projects/fastqc/) were used for quality control of raw sequencing data, and the sequence adapter was trimmed. Bowtie2[53] (v2.4.1) was used to remove prealignment reads (the mitochondrial genome, human alpha satellite repeats, human Alu repeats and human ribosomal DNA repeats) with parameters "-k 1 -D 20 -R 3 -N 1 -L 20 -I S,1,0.50 -X 2000 --rg-id". Then, parameters "--very-sensitive -X 2000 --rg-id" was used to align reads to the reference genome of human (hg19). Uniquely mapped reads were extracted by SAMtools[54] (v1.7) and marked duplicate with MarkDuplicates in Picard (v2.22.9, http://broadinstitute.github.io/picard). SAMtools was used to merge bam files of technical replicates for each sample. MACS2[55] (v2.2.6) was used to call accessible chromatin regions (ACRs) with parameters "-f BAM -g hs --nomodel --shift 100 --extsize 200 -B -q 0.05 --nolambda --SPMR --call-summits".

## ATAC-seq quality control

To ensure high quality of ATAC-seq data, we performed quality control at each analytical level. Profiles of four samples (A424R, A485R, A429R and A357R) were not included in analysis as the percentage of living cells was less than 10% and the percentage of mapping reads was less than 20,000,000. A total of 140 profiles from 75 samples of 59 patients were included for further analysis.

## Combination of ACRs on different levels

We extended 250 bp upstream and downstream from peak summit to get the ACR. ACRs were combined as previously reported[19] to get the sample level ACR. Briefly, ACRs were sorted by significance [$-\log_{10}(p$ value)]. For the overlapped ACRs, only the most significant ACR was kept. ACRs of diagnosis and relapse sample from the same patient were further combined for the patient level ACRs (p-ACR). Briefly, an ACR score [$-\log_{10}(p$ value)] was calculated for each sample level ACRs

and normalized by "score per million" in each sample. Two normalized ACR sets were then combined and re-sorted by the normalized ACR scores. Then for each most significant ACR, any less significant ACRs that overlapped with it were removed, resulting with the most significant ACRs as patient level ACRs. For the 13 diagnosis-only patients and 30 relapse-only patients, the sample level ACRs were taken as patient level ACRs. For the cohort level ACRs, the p-ACR of 59 patients were normalized individually and combined following a same procedure as described above. The final 758,738 ACRs were B-ALL ACRs on cohort level (c-ACR). In order to estimate the ACR expression in each sample, bam files were converted to bed files, and the read coverage for each ACR in each sample were calculated by BEDTools[56] (v2.29.2). The count of final 758,738 ACRs in 75 samples were used to calculate the CPM (count-per-million) by edgeR package[57] (v3.32.1). Only ACRs with $\log_2(CPM) > 0$ in at least 2 samples were kept for further analysis ($n = 625,287$).

## Annotation of accessible chromatin regions

ChIP-seq data of six histone modification markers (H3K4me1, H3K4me3, H3K9me3, H3K27ac, H3K27me3 and H3K36me3) and corresponding input data were downloaded from Blueprint Epigenome Consortium (http://blueprint-data.bsc.es/#!/, Donor ID: S017E3). FastQ Screen (v0.13.0) and FastQC (v0.11.9) was used for quality control of raw sequencing data. Burrows-Wheeler Aligner (v0.7.17-r1188) was used for mapping the reads to human genome (hg19). Uniquely mapped reads were extracted and PCR replicates were removed in bam files by SAMtools[54] (v1.7). ChromHMM states[21] was used to annotate whole genome into different states. Firstly, bam files were used as input for BinarizeBam function with parameters: "-f 2 -t outputsignaldir -b 200". The output signals were then used as input for LearnModel function, with 18 chromatin states. Finally, the chromatin states of S017E3 were combined into eight states, including TssA, Enh, BivR, Tx, Het, ZNF/Rpts, ReprPC and Quies. Accessible chromatin regions in 75 B-ALL samples were annotated with 8 chromatin states acquired above by BEDTools[56] (v2.29.2).

## Analysis of chromatin regions with differential accessibility in B-ALL

For the analysis for the chromatin regions with higher accessibility in B-ALL compared to B-progenitor cells, we combined ACRs of 75 B-ALL samples, 3 pre-pro B cells and 3 pro B cells and got the merged 750,197 ACRs for these 81 samples. A total of 643,274 recurrent ACRs (normalized $\log_2(CPM) > 0$ in at least two samples) were extracted from the merged ACRs and used to calculate differential accessible regions between B-ALL (75 samples) and B progenitor cells (6 cells) by DEseq2[58] (v1.30.1).

## ChIP-seq data processing

FastQ Screen (v0.13.0) and FastQC (v0.11.9) was used for quality control of raw sequencing data, and sequence adapter was trimmed. Burrows-Wheeler Aligner (v0.7.17-r1188) was used for mapping the clean reads to human genome (hg19). Uniquely mapped reads were extracted and marked duplicates with MarkDuplicates function in Picard (v2.22.9, http://broadinstitute.github.io/picard). MACS2 (v2.2.6) was used to call H3K27Ac modified regions of each sample with parameters "-f BAMPE -g hs -B".

## Analysis of Quies ACRs combining ChIP-seq data

To explore the biological function of Quies ACRs detected in B-ALL, we classified Quies ACRs into two groups. ACRs overlapped with genes (extended 5% of gene length in both upstream and downstream) were considered as gene regions, and all other ACRs were in distal regions. For the Quies ACRs within gene regions, we combined H3K27ac ChIP-seq signal to identify genes overlapped with both Quies ACRs and H3K27ac modification.

## Subtype-specific ACRs identification and transcription factor motif analysis

Subtype-specific ACRs were defined with following criteria: (1) recurrent ACRs with normalized $\log_2(CPM) > 1$ in more than 50% samples of the enriched subtype; (2) normalized $\log_2(CPM) > 1$ in less than 10% samples of all other subtypes. The transcription factor motif analysis was carried out with 101 bp center around ACR summit. The DNA sequence was extracted with getfasta function in BEDTools (v2.29.2). FIMO[59] (v 5.0.5) was used to scan for enriched transcription factor motif (FDR < 0.05). Only transcription factors with normalized $\log_2(FPKM) > 1$ in at least 1 sample of enriched subtype was included for further analysis.

## Analysis of subtype-enriched transcription factor and target gene expression

The percentage of subtype-specific ACRs with specific binding motif of each TF were calculated in individual B-ALL subtype to generate the heatmap in Fig. 2e. The subtype-enriched TFs were classified into nine groups based on the cluster result. In addition, TFs' expression were compared between B-ALLs in classified groups versus all other groups. For the analysis of target gene expression, genes with TSS within ±1 kb of subtype-specific ACRs with the TF binding motif were considered as potential target genes of the TF, and target gene expression in B-ALLs within classified groups versus all other groups were compared. The geometric average of $p$ values from all target genes were calculated to present the statistic difference of target genes for the TF. And the final fold change was median $\log_2$(fold change) of all target genes. $p$ values of TFs expression and target gene expression were calculated with one side Wilcox test.

## Allele-specific open chromatin analysis

The balanced transcription model was adapted from our previously published allele-specific expression identification model cis-X[60]. The model was optimized for chromatin accessibility analysis by integrating chromatin accessibility data and whole genome sequencing data. To estimate the adapted sigma in Gaussian distribution, we re-trained the parameters using 10 diagnosis samples, and determined the following function to estimate the adapted sigma: $\sigma(N) = 10.8(1 - e^{-\frac{N}{83}})$, where $N$ denotes the coverage at the tested genomic position. Firstly, genomic balanced SNP sites from WGS analysis with MAF between 0.3–0.7 were extracted. Secondly, these SNP sites were further filtered by ATAC-seq data, only SNP sites within ACRs with coverage ≥8 and alternative reads ≥3 were included. Thirdly, the accessible signal between two alleles on these SNP sites was calculated to identify imbalanced SNP sites ($p$ value < 0.05 and absolute delta ≥0.2) in ACRs. The $p$ values and delta values of all SNPs resided in each ACR were combined to score each ACR. ACRs satisfied $p$ value < 0.05 and absolute delta ≥0.2 were considered allelic imbalanced open chromatin regions. Cosmic genes with FPKM ≥ 1 and located within ±200 kb from peak of allele-specific open chromatin regions were considered as potential target genes.

## RNA-seq data processing

FastQ Screen (v0.13.0) and FastQC (v0.11.9) were used for quality control of raw sequencing data. STAR[61] (v2.7.1a) was used for mapping the clean reads to human genome (hg19). HTSeq count[62] (v0.11.2) was used to calculate reads located in each gene. Fragments Per Kilobase of exon model per Million mapped fragments (FPKM) was calculated for gene transcription quantification. CICERO[63] was used for fusion analysis. The copy number alterations and gene fusions from RNA-seq data were analyzed by RNAseqCNV[64] and Arriba[65] respectively.

## Association analysis linking ACRs to the potential target genes

To reveal the potential correlation between chromatin accessibility and gene expression, we predicted ACR-to-gene links in 52 samples

with both ATAC-seq and stranded RNA-seq data. For ACRs used for prediction, recurrent ACRs ($\log_2(CPM) > 0$ in at least 2 patients) with top 75% variance were extracted. Batch effect in RNA-seq data was corrected followed by renormalization across 52 samples, and only genes with top 75% variance among 52 samples were remained. The R package Matrix eQTL[66] (v2.3) was used to calculate the correlation between the expression of genes and ACRs from ATAC-seq. Only cis regulations were calculated in this analysis with ACR and gene located within 0.5 Mb on the same chromosome. ACR-gene associations with |beta| > 0.2 and FDR < 0.05 were kept in further analysis.

### Relapse-related ACRs in B-ALL and association with drug treatments

Differential ACRs between diagnosis and relapsed B-ALLs were calculated by DESeq2[58] (v1.30.1). Genes regulated by these relapse-related ACRs were predicted with ACR-to-gene links. Genes in response to B-ALL treatments were established by analyzing drug sensitivity (Area Under Curve, AUC) and gene expression data of 11 B-ALL cell lines (697, JM1, KASUMI2, MHHCALL3, MHHCALL4, NALM6, RCHACV, REH, RS4;11, SEM, SUPB15) from the DepMap database (https://depmap.org/portal/). Pearson correlation was calculated between gene expression and the AUC values for 8 drugs (Cyclophosphamide, Cytarabine, Dasatinib, Dexamethasone, Doxorubicin, Etoposide, Imatinib and Methotrexate). For drug-gene pairs with |correlation coefficient| > 0.5 and $p$ value < 0.05 were collected as drug related gene sets. Genes regulated by relapse-related ACRs were enriched for drug related gene sets by Fisher exact test ($p$ value < 0.05).

### Survival analysis

For comparison of OS and EFS between two treatment protocols (ALL-SCMC-2009 protocol and ALL-SCMC-2015 protocol), patients 118, 228, 273 and 284 with incomplete follow-up information and patients 155, 213, 289, 350 treated with ALL-SCMC-2005 protocol were excluded and the remaining 53 patients were included for analysis. For analysis of RFS-related ACRs, among all 75 samples of 59 patients, the samples A155R, A213R, A289R, A350R collected from four patients 155, 213, 289, 350 treated with ALL-SCMC-2005 protocol were excluded and the remaining 55 patients (29 diagnosis samples and 42 relapse samples) were analyzed. For each recurrent ACR, we divided these 42 relapse samples into two groups according to the median CPM. RFS (relapse-free survival) was defined from diagnosis to the first relapse event. Log-rank test was performed to estimate the difference in RFS rate between the two groups using Survival packages (v3.2-3) in R (v4.0.2).

### Reporting summary

Further information on research design is available in the Nature Portfolio Reporting Summary linked to this article.

## Data availability

All analysis in this study use reference genome of human (hg19) (https://ftp.ncbi.nlm.nih.gov/genomes/archive/old_genbank/Eukaryotes/vertebrate_mammals/Homo_sapiens/GRCh37/special_requests/). The raw ATAC-seq, RNA-seq and ChIP-seq data generated in this study have been deposited in the Genome Sequence Archive for Human (GSA-human) of the National Genomics Data Center of China under accession number HRA002815. The data are available for academic use under controlled access in compliance with the regulation of the Ministry of Science and Technology (MOST) of China for the deposit and use of human genomic data. Access can be obtained by contacting members of the Data Access Committee (DAC) following the application procedure in GSA. For detailed guidance, see GSA-Human_Request_Guide_for_Users (https://ngdc.cncb.ac.cn/gsa-human/document/GSA-Human_Request_Guide_for_Users_us.pdf). Data will be available immediately once the application was approved. The access to the

controlled data will be valid for 1 year from the date approved. The WGS data for 32 B-ALL patients and RNA-seq data for 29 B-ALLs were collected from previously published data[5]. Among these published data, the RNA-seq data were available in GSA-human of the National Genomics Data Center of China under accession number HRA000119, the processed genomic aberrations from WGS data used in this study were obtained from authors of the published paper[5] with raw data available in GSA-human under accession number HRA005668. The publicly available ATAC-seq data of 3 pre-pro B cells and 3 pro B cells were available in the National Center for Biotechnology Information's Gene Expression Omnibus with accession number GSE122989. The hyperdiploidy B-ALL cases of TARGET dataset were downloaded from Target website (dbGaP Sub-study ID phs000464) (https://gdc.cancer.gov/about-data/publications#/?groups=TARGET-ALL-P2&years=&order=desc). Only 43 samples with definitive molecular evidence for hyperdiploidy subtype were included in this analysis from TARGET dataset. ChIP-seq data of 6 histone modification markers (H3K4me1, H3K4me3, H3K9me3, H3K27ac, H3K27me3 and H3K36me3) were collected from Blueprint Epigenome Consortium (Donor ID: S017E3) (https://epigenomesportal.ca/ihec/grid.html?build=2020-10&assembly=4&institutions=3) and corresponding input raw data were downloaded from EGA database under accession number EGAD00001002421. The RNA expression data (DepMap Public 21Q1) and drug responses (Drug sensitivity AUC (CTD^2)) of 11 B-ALL cell lines were downloaded from DepMap database (https://depmap.org/portal/download). The COSMIC genes (release v87) were download from COSMIC database (https://cancer.sanger.ac.uk/cosmic/download).

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

## Acknowledgements

This work was supported by the National Natural Science Foundation of China (31970627 to Y.L.), Major Scientific Research Program for Young and Middle-aged Health Professionals of Fujian Province (2022ZQNZD011 to Y.L.), National Natural Science Foundation of China (82293660/82293665 to Y.L.), Shanghai Key Laboratory of Clinical Molecular Diagnostics for Pediatrics (20dz2260900 to Y.L.), Foundation of National Research Center for Translational Medicine at Shanghai (NRCTM(SH)-2019-04 to S.S.) and Shanghai Committee of Science and Technology (21ZR1441000 to F.Y.).

## Author contributions

Y.L. and S.S. designed and supervised the study. H.W., F.Z., F.Y. and R.W. performed the experiments. H.S., H.W. and B.L. analyzed genomic data with the help from B.C., J.R., S.Z., W.W., J.L., X.C. and K.W.. L.D., X.W., Y.T., W.H., J.C., Y.X., B.L. and J.T. collected clinical samples and information. Y.L., H.W. and H.S. wrote the manuscript.

## Competing interests

The authors declare no competing interests.
