## [Peer Review File · Nature Communications]

Chromatin accessibility landscape of relapsed pediatric B-lineage acute lymphoblastic leukemiaReviewers' Comments:

Reviewer #1:

Remarks to the Author:

Wang H and coworkers describe a cohort of 61 pediatric patients with relapsed B-ALL which have been profiled for accessible chromatin regions (ACRs) by ATAC-seq. Complementary omics analyses were performed in selected cases (WGS, RNA-Seq, H3K27ac CHIP-Seq). A subset of patients was represented with multiple sampling timepoints in the cohort.

The authors used comparison to a published ATAC-seq dataset of mature B cells to identify B-ALL specific ACRs, showing an overall more accessible chromatin in B-ALL with enrichment of promotor regions of genes related to cell cycle and leukemogenesis. Subtype-specific sets of ACRs were defined for the 5 largest molecular B-ALL subtypes represented in the cohort. Signatures of associated transcription factors were validated by RNA-Seq. Further analysis suggested an involvement of Quies chromatin regions in B-ALL transcriptional regulation, as exemplified for a B-ALL specific regulation of the MYC-enhancer cluster (BENC). Integration of WGS and ATAC-seq data was used to define allele-specific open chromatin regions, identifying allele-specific transcription in oncogenes. Substantial heterogeneity in ACR gains and losses at relapse was observed in a limited number of paired samples with gene sets related to B-ALL in-vitro treatment response showing enrichment in relapse associated ACRs. Unsupervised analysis (t-SNE) of relapse-specific ACRs in patient subset with available data (n=42) identified two clusters enriched for distinct B-ALL subtypes with also distinct outcomes. The authors focus on 3 hyperdiploid B-ALL cases clustering with the poor prognosis cases and identify an associated gene signature which could be reproduced in 3 poor responding Hyperdiploid B-ALL cases in an independent cohort (n=29 hyperdiploid B-ALL).

The study is technically robust and well described. To the best of our knowledge, it represents the largest ATAC-seq analysis of B-ALL currently available. A similar approach has currently been pre-published (bioRxiv 2023.02.14.528493; doi: <https://doi.org/10.1101/2023.02.14.528493>). This largely descriptive analysis provides a valuable resource of accessible chromatin as a new omics layer in the characterization of the B-ALL genomic landscape with definitions of subtype-specific epigenetic regulations. Conclusions on the B-ALL specific regulations are somewhat limited due to comparison to mature B cells (n=4 donors). Results presented as 'B-ALL specific' could therefore be at least partially specific to B lymphoid progenitors as well. Conclusions on clinical implications – especially regarding the identification of a high-risk hyperdiploid B-ALL subset - are of high interest, however the small sample size precludes any definite conclusion here. Several concerns regarding the methodology and conclusions of the manuscript warrant further attention.

MAJOR POINTS:

- 1.) Some Patients are represented by multiple samples (technical replicates, diagnosis / relapse pairs) in the study. Further clarification is warranted how the authors avoided potential bias in the selection of specific ACR signatures introduced by the inclusion of multiple samples per patient. How were multiple samples per patient used for the different analyses presented?
- 2.) What was the reason to include diagnosis only samples, especially in ChIP-seq where these represent more than half of the patients analyzed?
- 3.) What was the basis of patient selection? In which way are these patients 'high-risk' as stated in the manuscript title? Is this selection representative of any clinical situation? What was the reason to include relapsed cases together with first diagnosis cases?
- 4.) Defining B-ALL specific ACRs based on comparison to mature B-cells leaves open which of these ACRs might rather represent the B progenitor origin of B-ALL. Especially the observation that B-ALL has a more transcriptionally active chromatin structure might be true of B progenitors as well. These

analyses would highly benefit from direct comparison to B progenitors. Otherwise, this should be discussed in the manuscript and the corresponding conclusions should be toned down.

5.) t-SNE analyses use for definition of patient clusters depend strongly on the hyperparameters chosen. Is there any independent validation to support especially the clusters in Figure 5A?

6.) What is the karyotype of the poor-risk Hyperdiploid cases? If conventional analyses are not available, maybe virtual karyotyping could be performed to determine if it matches to established patterns or shows specific characteristics potentially contributing to the clinical phenotype.

7.) In which way do the clusters identified in Figure 5A represent distinct prognoses and time to relapse (manuscript line 306)? Are these correlations independent of the aggregation of molecular subtypes in these clusters?

8.) The number of patients representing cluster 1,2 and 3 should be included in Figure 6F. It seems that only n=3 patients represent cluster 3. If so, Kaplan Meier plots bear a high risk of overestimating the robustness of this data and should therefore be removed.

MINOR POINTS:

1.) It is not clear how many patients were included in analyses for Figure 5F. Figure legend says n=43, manuscript text says n=43 samples (n=14 of these diagnosis-relapse paired). Please clarify. Was clustering of the patients stable also when only one sample w per patient as used)

2.) Color codes in Figure 5A are difficult to read.

3.) The authors might want to consider use of current HUGO / WHO / ICC nomenclature for abbreviation of molecular subtypes (e.g., TCF3::PBX1 or KMT2A instead of E2A or MLL) to make figures more easily accessible.

Reviewer #2:

Remarks to the Author:

The authors of this paper have explored the chromatin accessibility landscape of 11 different genetically defined subtypes of high-risk B lineage ALL by comparing leukemic cells of relapsed ALL patients with normal B cells obtained from human cord blood. The authors find that enrichment of accessible chromatin regions in leukemia. These ACRs are located in areas that do not contain histone modifications, which are characteristic of transcriptional activity in normal B cells. Indeed, most of the ACR in the leukemia cells also contained H3K27ac modifications suggesting the presence of transcriptional activity.

The authors further show that the different subtypes exhibited distinguishable profiles of ACRs. Transcription factor motif analysis showed a certain correlation with the transcription of the respective target genes. The association of ACRs with leukemia related single nucleotide polymorphisms suggest that these ACRs might be related to activity of neighboring genes. However, transcriptional activity is directly documented in only two of such genes.

When comparing ACRs and transcriptional profiles of samples obtained either at diagnosis or at the time of relapse the authors identify an association of a cluster ACRs and deregulated target genes with relapse free survival. Relapse-free patients thus separate from those experiencing a relapse, which mostly correlates with known genetic subtypes associated with favorable and unfavorable prognosis, respectively. In this analysis, 3 hyperdiploid patients (a subgroup that is generally associated with a favorable risk profile) who developed a relapse were correctly clustered together with the other high-risk patients who also developed a relapse. These leukemias were found to mimic an expression profile

of more immature cells and of myeloid progenitors suggesting the presence of known high-risk features, which are not commonly associated with hyperdiploid precursor B-ALL. This expression profile was also found in relapsing patients in another set of hyperdiploid patients documented in the TARGET database.

In sum, the data presented here describe features of chromatin accessibility in precursor B-ALL in comparison to a heterogeneous population of normal B-cells. This part of the paper suffers from the poor definition of the normal cells. In particular, it is known that ALL cells are derived from immature precursors. Therefore, it is critical to compare leukemic cells with normal early B-cell precursors. The reference B cells obtained from cord blood are likely a mixture of some immature and predominantly mature normal cells. Any differences between leukemic cells and such an ill-defined mixture of normal cells might thus represent differences between mature and immature cells and not necessarily between leukemic cells and their normal counterpart of comparable immaturity.

By contrast, the comparative data of diagnostic and relapse samples are potentially of more substantial interest. The identification of a high-risk expression and ACR profile in hyperdiploid precursor B-ALL even at the time of initial diagnosis is novel and might be clinically useful as biomarker indicating risk. However, unequivocally documenting this usefulness would require the analysis of a larger clinically well annotated group of unselected patients rather than the 43 hyperdiploid samples extracted from the TARGET database.

Reviewer #3:

Remarks to the Author:

In the current manuscript, the authors Wang et al, have generated chromatin accessibility profiles on 79 paediatric B-ALL tumours from a total of 61 patients. This was combined with WGS and RNA-seq for a subset of patients and H3K27Ac ChIP-seq on another subset of tumours. These patients represented 11 B-ALL subtypes. This is an interesting data set to have for the field, but the paper suffers from being overly descriptive and limited novelty. The paper would have benefitted from functional experiments to support some of the interesting findings.

Major comments:

1) I have some concerns on the strategy used to purify tumour cells, those with low tumour count % according to the Table S2. Some of these are in the range of 20-40% only and yet it appears that the cells were sorted based on CD19+/CD10+. Some patients have no proportions. How can the authors be confident that their strategy was sufficient to purify leukaemia cells? Albeit at diagnosis/relapse it is likely that the burden is very high. Furthermore, whilst they show two representative flow cytometry plots - patient A442 does not exist in the Table S2?

2) Landscape assignment for ACR's. I have a major concern on the authors partitioning of the genome into functional regions using primary B-cells as a reference. The genomic regions and their assignment require H3K27Ac, H3K4me1, H3K4me3 and H3K27me3 - none of which were carried out on these samples and so the claims that ACR's are in "enhancer" or "Quiescent" regions are misplaced. Similarly, to say the repressed regions has less accessibility is misleading, as again - the repressed region is in the normal B-cell. It might not be repressed in B-ALL and the histone modifications are likely to be very different between the different subtypes of B-ALL at different regions. In the absence of the authors undertaking the appropriate ChIP-seq experiments, this analysis and conclusions are limited.

3). Transcription factor motif analysis: The authors have analysed the ACR's and clustered them according to a predefined set of 110 different transcription factors. There is some nice TF sub-type specificity. However, the authors then seem to cherry pick E2F6 and THAP1 to show the relevance of this analysis and ignore all the others in the list that are enriched for each subtype. Can the authors look more globally at each motif and determine which TF's might be relevant based on expression?

Moreover, they suggest overlap between MLL rearranged and ZNF-384 subtypes, and yet there is greater overlap between E2A and ETV subtypes which they do not describe. Can the authors comment? The authors also claim in the figure 2 legend for panel E that this clustering is based on occupancies of the TF's. This is not the case - this is motif enrichment as no ChIP-seq was undertaken for all 110 TF's to support this claim.

4) Analysis of Diagnosis and Relapse: Relapse remains an important and significant clinical challenge and epigenetic and genetic changes have been implicated. The authors integrate drug susceptibility with 1259 genes that are potentially deregulated in relapse. There is an association between Dasatinib and PH/PHL B-ALL. The authors then claim that treatment could reshape the chromatin. However, there may have also been clonal selection. Have the authors looked at clonal composition and outgrowth of clones with different mutations?

Point-by-point Response to Reviewers' Comments

Reviewer #1, expertise in pediatric high-risk B-ALL genomics (Remarks to the Author):

Wang H and coworkers describe a cohort of 61 pediatric patients with relapsed B-ALL which have been profiled for accessible chromatin regions (ACRs) by ATAC-seq. Complementary omics analyses were performed in selected cases (WGS, RNA-Seq, H3K27ac CHIP-Seq). A subset of patients was represented with multiple sampling timepoints in the cohort.

The authors used comparison to a published ATAC-seq dataset of mature B cells to identify B-ALL specific ACRs, showing an overall more accessible chromatin in B-ALL with enrichment of promotor regions of genes related to cell cycle and leukemogenesis. Subtype-specific sets of ACRs were defined for the 5 largest molecular B-ALL subtypes represented in the cohort. Signatures of associated transcription factors were validated by RNA-Seq. Further analysis suggested an involvement of Quies chromatin regions in B-ALL transcriptional regulation, as exemplified for a B-ALL specific regulation of the MYC-enhancer cluster (BENC). Integration of WGS and ATAC-seq data was used to define allele-specific open chromatin regions, identifying allele-specific transcription in oncogenes. Substantial heterogeneity in ACR gains and losses at relapse was observed in a limited number of paired samples with gene sets related to B-ALL in-vitro treatment response showing enrichment in relapse associated ACRs. Unsupervised analysis (t-SNE) of relapse-specific ACRs in patient subset with available data (n=42) identified two clusters enriched for distinct B-ALL subtypes with also distinct outcomes. The authors focus on 3 hyperdiploid B-ALL cases clustering with the poor prognosis cases and identify an associated gene signature which could be reproduced in 3 poor responding Hyperdiploid B-ALL cases in an independent cohort (n=29 hyperdiploid B-ALL).

The study is technically robust and well described. To the best of our knowledge, it represents the largest ATAC-seq analysis of B-ALL currently available. A similar approach has currently been pre-published (bioRxiv 2023.02.14.528493; doi:<https://doi.org/10.1101/2023.02.14.528493>). This largely descriptive analysis provides a valuable resource of accessible chromatin as a new omics layer in the characterization of the B-ALL genomic landscape with definitions of subtype-specific epigenetic regulations. Conclusions on the B-ALL specific regulations are somewhat limited due to comparison to mature B cells (n=4 donors). Results

presented as 'B-ALL specific' could therefore be at least partially specific to B lymphoid progenitors as well. Conclusions on clinical implications – especially regarding the identification of a high-risk hyperdiploid B-ALL subset - are of high interest, however the small sample size precludes any definite conclusion here. Several concerns regarding the methodology and conclusions of the manuscript warrant further attention.

MAJOR POINTS:

1.) **Reviewer1:** *Some Patients are represented by multiple samples (technical replicates, diagnosis / relapse pairs) in the study. Further clarification is warranted how the authors avoided potential bias in the selection of specific ACR signatures introduced by the inclusion of multiple samples per patient. How were multiple samples per patient used for the different analyses presented?*

[Author's response] Thank you for the comments. As the reviewer pointed out, we made two technical replicates for 65 B-ALL tumor samples with sufficient material. This experiment served as quality control of ATAC-seq experiment. As showed in Supplementary Figure 1C and 1D in the original manuscript, we observed high correlation between the replicates for all samples. Data from replicates collected from the same sample were combined for downstream analysis. The peaks were processed into sample level accessible chromatin regions (ACRs) as previously described (Corces et al., Science 2018, PMID: 30361341). Briefly, the ACRs were sorted by significance [$-\log_{10}(\text{P value})$] and only the most significant one was kept for the overlapped ACRs ("Combination of ACRs on different levels" in Methods section).

Among the samples analyzed with ATAC-seq in current study, we had 29 diagnosis tumor samples and 50 relapsed tumor samples collected from 61 relapsed B-ALL patients. The diagnosis and relapse samples were analyzed separately for the sample level ACRs. Then the sample level ACRs from diagnosis and relapsed samples of the same patient were combined into patient level ACRs (p-ACRs), and the p-ACRs were further combined into the cohort level ACRs (c-ACRs, "Combination of ACRs on different levels" in Methods section). The c-ACRs were used as the basis for representing the accessible chromatin regions in B-ALL genome in this study and quantified in individual sample for the downstream analysis. We documented in below table the use of ACRs in each analysis.

Analysis	Type of ACRs used	Type of Samples used
1 Chromatin accessibility landscape in B-ALL	Sample level ACRs and cohort level ACRs (c-ACRs)	Both diagnosis and relapsed samples were included.
2 Subtype-specific ACRs	c-ACRs quantified in each sample	Both diagnosis and relapse samples were included.
3 Analysis of allelic-specific open chromatin (ASOC)	Sample level ACRs	One sample was used to represent the patient in evaluating the ASOC regions across the genome to avoid potential over-representation of ACOS regions in patients with multiple samples. For the diagnosis-relapsed paired samples, the diagnosis sample was selected. For patients with only diagnosis or relapsed sample, the available sample was used. All the diagnosis and relapsed samples were included when summarizing the leukemia associated SNPs in Figure 3D.
4 ACR changes between diagnosis and relapsed B-ALL	c-ACRs quantified in each sample	Both diagnosis and relapsed samples were included. Differential ACRs were analyzed by comparing between diagnosis and relapsed samples.
5 RFS associated ACRs	c-ACRs quantified in each sample	Only relapsed samples were included to calculate the RFS associated ACRs. Diagnosis samples were included in the clustering analysis which was based on the RFS associated ACRs identified.

As showed above, there were two analyses involving sample selection. One was during the ASOC analysis. In this analysis, we only included the diagnosis samples if both diagnosis and relapsed samples were available for the same patient when evaluating the ASOC pattern across the genome. We did this to avoid potential overestimation of some genomic regions due to the inclusion of both diagnosis and relapsed samples. During the revision, we repeat this analysis to include all the samples. As showed in the Extended Figure 1 below, both the

percentage of genome showed ASOC signature and the genome partitioning of the ASOC regions in the genome were not changed. To improve the clarity of our manuscript, we added a column showing the patient number in revised Figure 3D (attached below) and the description of sample usage in the legend of Figure 3 and Supplementary Figure 6.

Extended Figure 1. The allelic-specific open chromatin (ASOC) regions in B-ALL with both diagnosis and relapsed B-ALL samples included in this analysis, as compared to the results showed in Figure 3A-B and Supplementary Figure 6A-B.

SNP ID	Position	Ref allele	Alt allele	Biased allele	Sample No.	Patient No.
rs7290465	chr22:50679758	A	G	A	15	13
rs7090445	chr10:63721176	C	T	T	14	11
rs7241766	chr18:13517327	T	C	T	8	8
rs13401811	chr2:111616104	G	A	G	8	6
rs674313	chr6:32578082	C	T	T	7	7
rs1342854	chr9:5791151	G	A	G	7	5
rs539846	chr15:40397936	G	T	G	7	5

Figure 3D (revised). The table displays detailed information about the seven SNPs shown in Figure 3C.

The other analysis involving sample selection was RFS associated ACRs analysis. We included only relapsed sample for each patient. The aim of this analysis was to identify ACR features associated with higher risk of early relapse (relapse time). Following this, the relapsed tumor samples better represented the chromatin accessibility state of tumor genomes with different relapse time. We identified 70,573 RFS associated ACRs in this analysis. With these ACRs from analyzing relapsed tumor samples, we further asked if these features were already existed in the diagnosis samples. The results did show a similar pattern of ACR features in the matched diagnosis samples (Supplementary Figure 7C of the revised manuscript), supporting this hypothesis.

2.) **Reviewer1:** *What was the reason to include diagnosis only samples, especially in ChIP-seq where these represent more than half of the patients analyzed?*

[Author's response] All patients included in this study were relapsed. We collected the paired diagnosis and relapsed tumors for the experiments. However, the ATAC-seq and ChIP-seq data were missing for some samples due to unavailable of high viability tumor cells. In this case, there will be diagnosis or relapsed only samples analyzed. The diagnosis only tumors, as pointed out by the reviewer, were collected from patients who relapsed during treatment. No further selection of samples or patients were carried out in this experiment. We included these diagnosis samples to represent the chromatin accessibility of relapsed B-ALLs before treatment. The H3K27ac ChIP-seq data were included as a supplementary to the public available data to investigate the potential functions of ACRs in B-ALL tumor genome.

3.) **Reviewer1:** *What was the basis of patient selection? In which way are these patients 'high-risk' as stated in the manuscript title? Is this selection representative of any clinical situation? What was the reason to include relapsed cases together with first diagnosis cases?*

[Author's response] In current study, we recruited relapsed B-ALL patients treated at Shanghai Children's Medical Center through 2007 to 2019. The diagnosis and relapsed tumor samples with adequate material were collected and analyzed. No further selection of patient was applied. The 'high-risk' in the manuscript referring to patients with relapsed B-ALL. As mentioned in the

introduction of the manuscript, relapse is one of the major challenges in B-ALL treatment. Although the overall survival of pediatric B-ALL is over 90%, outcome for relapsed B-ALL patients remained poor. With the advances in understanding the genomic aberrations behind this malignancy, patients were stratified into low-risk, intermediate-risk and high-risk and received different therapy accordingly in clinic. As all the patients included in this study were relapsed B-ALLs, we used the 'high-risk' in the original manuscript. We realized the use of 'high-risk' was not appropriate and introduced confusion about the patient selection. We have changed the 'high-risk' to 'relapsed' in the revised manuscript.

The inclusion of both diagnosis and relapsed tumor samples collected from relapsed B-ALL allowed us to find the difference of chromatin accessibility during the treatment and investigate the potential mechanisms for B-ALL relapse from chromatin accessibility aspect. We identified a median of 945 ACRs (ranging from 268 to 4,072) showed significant different accessibility between diagnosis and relapsed tumors in major B-ALL subtypes ("Chromatin accessibility changes in response to B-ALL treatment" in Results section). Besides, we observed a high heterogeneity in chromatin accessibility changes during relapse. These results allowed us to understand the changes during B-ALL relapse from chromatin accessibility.

4.) **Reviewer1:** *Defining B-ALL specific ACRs based on comparison to mature B-cells leaves open which of these ACRs might rather represent the B progenitor origin of B-ALL. Especially the observation that B-ALL has a more transcriptionally active chromatin structure might be true of B progenitors as well. These analyses would highly benefit from direct comparison to B progenitors. Otherwise, this should be discussed in the manuscript and the corresponding conclusions should be toned down.*

[Author's response] We thanks the reviewer to point out this important question. In this revision, we re-analyze the data to directly compare the chromatin accessibility of B-ALL to progenitor B cells. We collected the public available ATAC-seq data for pre-pro B and pro B cells (O'Byrne et al., 2019. PMID: 31383639). Firstly, the pre-pro B and pro B cells showed comparable number of ACRs in individual genome as compared to B-ALLs, indicating the previously observed more accessible chromatin regions in B-ALLs is a difference between pre-pro B/pro B cell and mature B cells. We have modified the results part on page 6 to represent this change in the revised manuscript.

We next compare the ACRs detected in B-ALL to pre-pro B and pro B cells. Results showed that the majority of ACRs in pre-pro B cells (98.57%) and pro B cells (98.35%) were also observed in B-ALLs, further supporting the B progenitor origin of B-ALL. We included this part of results as a new figure panel of Figure 1C in the revised manuscript. In addition, we detected 585,248 ACRs showed specific accessibility in B-ALL, accounting for 78.39% of the total ACRs in B-ALL. High heterogeneity was observed in these B-ALL-specific ACRs as showed in the new figure panel in Figure 1D in the revised manuscript. We further identified a total of 252,028 recurrent ACRs which showed higher accessibility in B-ALL compared to B-cell progenitors. Enrichment analysis showed that these ACRs were associated with genes in tumor-associated biological processes including proliferation/differentiation, immune process, signal transduction and metabolic process (showed as a new figure panel in Figure 1E and new Supplementary Table 6). Among these ACRs with increased accessibility in B-ALLs were known oncogenes. We found increased chromatin accessibility in both the TSS regions of oncogenes (*IL7R*, *TCL1A*, *TCF3*, *RHOA* and *ELL*, among others, showed as a new figure panel in Figure 1F) and well-established enhancer regions regulating transcription of oncogene (distal blood enhancer cluster (BENC) of *MYC*, showed as revised figure panel in Figure 1G). These observations suggested that chromatin accessibility changes were involved in B-ALL. We believed the revised results more clearly reflected the chromatin accessibility changes in B-ALL.

We have modified the “B-ALL-specific chromatin accessible regions associated with leukemogenesis” paragraph in the results section on pages 6 to 7 in the revised manuscript. And the updated results were organized into revised Figure 1C-G, which was also attached below.

Figure 1C-G (revised). (C) Venn diagram shows the overlap between ACRs detected in B-ALL and B-cell progenitors (pre-pro B cells and pro B cells). (D) Violin plot presents the recurrence of ACRs in B-ALLs. The ACRs were grouped into four groups as showed in Figure 1C. Significant difference was observed among the four groups ($P < 0.0001$, Kruskal-Wallis test). (E) Gene set enrichment analysis shows that 2,332 protein coding genes regulated by 252,028 higher accessible ACRs in B-ALL were enriched in tumor associated biological processes (Supplementary Table 6). Only terms with $FDR < 0.001$ are displayed. The node size represents the enriched FDR values, and the edge represents overlap between two gene sets. Clusters of functionally related categories were manually grouped and labeled in different colors. (F) Wiggle plot shows regions with increased chromatin accessibility in B-ALLs as compared to B-cell progenitors. ACRs with +/- 1kb centered the TSS of representative cosmic genes are showed. Only subtypes with more than three cases are included and two samples are randomly selected and showed for each subtype. The ACR present higher accessible in B-ALL upstream TSS of *IL7R* are highlighted in light-yellow. (G) Wiggle plot shows the chromatin accessibility in the blood enhancer cluster (BENC) region (chr8:126,712,193 – 128,412,193). The positions of enhancers (A to I) are indicated in the BENC track on the top. The tracks showing ACRs in this region are organized as in Figure 1F.

Two enhancers from the BENC cluster showing increased accessibility in B-ALLs are highlighted in light-yellow.

5.) **Reviewer1:** *t-SNE analyses use for definition of patient clusters depend strongly on the hyperparameters chosen. Is there any independent validation to support especially the clusters in Figure 5A?*

[Author's response] Thank you for this comment. We believe the reviewer is referring to the clustering result in Figure 5B. We have repeated this analysis using the UMAP and PCA methods. Both analyses showed concordant results and separated the patients with different RFS status, as showed in a new figure panel in Supplementary Figure 7B (also attached below). The patients in Group B from all these analyses experienced early relapse as showed on the right column of Supplementary Figure 7B (with darker red color indicate shorter relapse time). Furthermore, we performed a new analysis to validate this clustering results in the independent TARGET B-ALL cohort. We identified a total of 10,975 RFS associated ACRs with higher accessibility in B-ALLs from Group B ($|\log_2FC| > 1$ and $FDR < 0.05$) and found 1,827 potential target genes from ACR-to-gene association analysis (as a new Supplementary Table 14). With these genes, 252 B-ALL samples from TARGET project were grouped into 3 clusters (as a new Supplementary Figure 7D and Supplementary Table 15). Survival analysis showed significant differences in both event-free survival (EFS) and overall survival (OS) between the clusters (as a new Supplementary Figure 7E). Patients showed the highest expression of the target genes were associated with the worst prognosis (Cluster 3). These results confirmed that genes associated with the differential accessible ACRs between the two patient groups in original Figure 5B could separate B-ALL patients from independent TARGET cohort with different prognosis, further validating the two patient clusters were prognosis associated.

We have included these results in the revised Supplementary Figure 7 (attached below) and modified relevant writings on page 11 in the revised manuscript.

Supplementary Figure 7 (revised). Chromatin accessibility is associated with the prognoses of B-ALL patients. (A) RFS (up) and OS (down) were estimates for B-ALL patients with sufficient follow-up information. No significant difference was observed between patients treated with the ALL-SCMC-2009 and ALL-SCMC-2015 protocols (Log-rank test). (B) The unsupervised clustering with 3 methods (t-SNE, UMAP and PCA) of 42 relapse B-ALL cases based on the top 10% most

significant RFS-related ACRs. Each dot represents an individual sample. The color in the left column plots represents subtype, and the color in the right column plots reflects the time to relapse. (C) The t-SNE plot of 42 relapse and 29 diagnosis samples based on the same RFS-related ACRs shown in Figure 5B. Each dot represents an individual sample, and color represents B-ALL subtype. (D) Unsupervised clustering analysis of 252 B-ALL samples from the TARGET project based on the expression of target genes potentially regulated by differential ACRs with increased accessibility in B-ALLs of Group B. (E) Event-free survival rate (EFS, 202 patients) and overall survival rate (OS, 201 patients) estimates for B-ALL samples of three clusters presented in Supplementary Figure 7D. Only patients with sufficient follow-up information were included. For patients with paired diagnosis and relapsed samples, the patient was assigned to the cluster of relapsed samples in the survival analysis. Log-rank test, EFS $P = 2.564e-28$ and OS $P = 5.038e-17$.

6.) **Reviewer1:** *What is the karyotype of the poor-risk Hyperdiploid cases? If conventional analyses are not available, maybe virtual karyotyping could be performed to determine if it matches to established patterns or shows specific characteristics potentially contributing to the clinical phenotype.*

[Author's response] The karyotype of patients analyzed in this study were collected in Supplementary Table 1. Data was not available for two out of three hyperdiploidy patients of Group B in Figure 5B. Instead, we turned to the copy number variation (CNV) results from WGS analysis as we previously reported (Li et al., 2020. PMID: 31697823). The results of all 6 relapsed hyperdiploidy samples with WGS data in Figure 5B were showed in the Extended Table 1 below. We performed Fisher exact test on each chromosome between Group A and Group B, and there was no significant difference observed. This was at least partially due to the small number of patients in current analysis and worth more investigation in the future.

Sample	Groups of Figure 5B	CNV from WGS
A118R	Group B	52,+6,+14,+17,+18,+21,+X
A174R	Group B	53,+6,+8,+14,+17,+18,+21,+X
A197R	Group A	55,+4,+5,+6,+10,+14,+17,+18,+21,+X
A228R	Group A	53,+4,+6,+8,+14,+18,+21,+X
A233R	Group B	48,+17,+21
A262R	Group A	55,+1,+4,+6,+7,+10,+14,+18,+21,+X

Extended Table 1. Arm level CNV of the 6 hyperdiploidy B-ALLs showed in Figure 5B. The CNV were resulted from analyzing whole genome sequencing data and extracted from Li et al., 2020. PMID: 31697823.

7.) **Reviewer1:** *In which way do the clusters identified in Figure 5A represent distinct Prognoses and time to relapse (manuscript line 306)? Are these correlations independent of the aggregation of molecular subtypes in these clusters?*

[Author's response] Patients in Group B of Figure 5B showed significant inferior RFS (p -value = $6.755e-10$) and OS (p -value = $2.097e-7$) as compared to patients in Group A. We have included the KM-plot showing this difference in the revised Figure 5C (attached below).

Figure 5C (revised). The Relapse-free survival rate (RFS) and overall survival rate (OS) estimates for B-ALL samples of Group A and Group B. (Log-rank test, $P = 3.778e-10$ for RFS and $P = 5.585e-7$ for OS).

For the B-ALL molecular subtypes, patients in Group A were enriched in the ETV6::RUNX1 and hyperdiploidy subtypes and patients in Group B were mostly KMT2A and BCR::ABL1/BCR::ABL1-like subtypes, as we described in “Chromatin accessibility features affect patient outcomes” in the results section of the manuscript on pages 11-12. Although patients from the specific molecular subtype tend to be grouped together in Figure 5B, we found minimal overlap between the RFS related ACRs and the subtype-specific ACRs, or with the most variable ACRs in B-ALLs which reflecting the variance among different molecular

subtypes (Extended Figure 2 below). These results supported that the ACRs separating the two clusters in Figure 5B were RFS associated but not subtype associated.

Extended Figure 2. Venn diagram shows the comparison of RFS associated ACRs, subtype-specific ACRs, and the most variant ACRs in B-ALLs which reflecting the variance among different molecular subtypes.

8.) **Reviewer1:** *The number of patients representing cluster 1,2 and 3 should be included in Figure 6F. It seems that only n=3 patients represent cluster 3. If so, Kaplan Meier plots bear a high risk of overestimating the robustness of this data and should therefore be removed.*

[Author's response] We thanks the reviewer for this suggestion. The numbers of patients were added in the revised manuscript. There were 5 tumor samples collected from 3 patients in cluster 3. We estimated the 95% confidence interval in the survival analysis. As showed in the revised Figure 5G (attached below), although the number of patients were limited, patients in cluster 3 showed significant difference as compared to patients in the other two clusters. All 3 patients relapsed significantly earlier than patients in the other clusters. We believe this cross-platform analysis in independent cohort could provide useful information to support our finding of prognosis associated ATAC-seq signature in hyperdiploidy B-ALL. That said, we agree with the reviewer that the statistical analysis suffered from the small number of patients in this analysis. We have

included this statement on page 12 to clearly state the limitation of this analysis and toned down of this claim in the revised manuscript.

Figure 5G (revised). Event-free survival rate (EFS, n=35) and overall survival rate (OS, n=36) estimates for hyperdiploidy B-ALL samples from the TARGET project. Cases in Cluster3 with high expression of the target genes and mimicking the hyperdiploidy B-ALL cases in Group B had worse outcomes. (Log-rank test, EFS P = 0.0012 and OS P = 0.0002).

MINOR POINTS:

1.) **Reviewer1:** *It is not clear how many patients were included in analyses for Figure 5F. Figure legend says n=43, manuscript text says n=43 samples (n=14 of these diagnosis-relapse paired). Please clarify. Was clustering of the patients stable also when only one sample w per patient as used)*

[Author's response] Thanks to point out this question. There is a typo in original manuscript. There were 7 patients (instead of 14) with both diagnosis and relapse tumor samples. The number of patients included in the clustering (Figure 5F in revised manuscript) and survival analysis (revised Figure 5G) were the same (36 patients). The clustering analysis is performed on tumor samples, with a total of 43 samples from the 36 patients were analyzed (including both diagnosis and relapse samples). Meanwhile, the prognosis analysis was on patients (a total of 36 patients included). And the EFS information is missing for one patient, resulting a total of 35 patients included for EFS analysis. We have included the detailed number of patients in the revised Figure 5G and modified the description in results and figure legends.

For the cluster of TARGET cohort in Figure 5F (Figure 5E in the original manuscript), we included both diagnosis and relapsed samples. As we showed in Supplementary Figure 7C, the RFS-associated ACR signatures were shared between diagnosis and relapsed tumors, suggesting this ACR signature was present in the diagnosis samples. As a fact, clustering results in Figure 5F showed that the diagnosis and relapsed tumors collected from the hyperdiploidy B-ALLs (PARMSP and PARJSR) predicted with high-risk for early relapse (Cluster 3) did group together. During the revision, we redo the clustering analysis with relapsed tumors only. There was only 18 relapsed hyperdiploidy tumors from TARGET project. As showed in the Extended Figure 3 below, the results could be largely replicated as two out of the three high-risk hyperdiploidy B-ALLs were still clustered together on the far left of the heatmap. One case did shift to another cluster, suggesting the robustness of this pattern was limited to the small number of patients and remained to be improved. We included this statement in the revised manuscript on page 12.

Extended Figure 3. Unsupervised clustering analysis of 18 relapsed hyperdiploidy samples from the TARGET project based on the expression of 603 target genes regulated by 3,156 ACRs up-regulated in Group B. The 3 relapse samples of patients grouped in Cluster 3 (high risk patients for early relapse) of Figure 5F were highlighted with red rectangle.

2.) **Reviewer1:** Color codes in Figure 5A are difficult to read.

[Author's response] Thank you for your comments. We have changed the color scales in the revised Figure 5A (attached below) to make it clearer.

Figure 5A (revised). Gene set enrichment analysis shows that 3,976 target genes regulated by 70,573 RFS-related ACRs are enriched for hematopoietic development and cell cycle-associated biological process terms. The categories with the most significant FDR values are listed and are sorted by FDR values in reverse order.

3.) **Reviewer1:** *The authors might want to consider use of current HUGO / WHO /ICC nomenclature for abbreviation of molecular subtypes (e.g., TCF3::PBX1 or KMT2A instead of E2A or MLL) to make figures more easily accessible.*

[Author's response] Thank you for this suggestion. We have updated the manuscript to be consistent with the current nomenclature. The detailed changes are listed below.

In original MS	Updated in revised MS
MLL	KMT2A
E2A	TCF3::PBX1
ETV	ETV6::RUNX1
HYPER	hyperdiploidy
HYPO	hypodiploidy
HLF	TCF3::HLF
PAX5	PAX5alt
PH	BCR::ABL1
PHL	BCR::ABL1-like
ZNF384	ZNF384
MEF2D	MEF2D

Reviewer #2, expertise in pediatric high-risk B-ALL epigenomics (Remarks to the Author):

The authors of this paper have explored the chromatin accessibility landscape of 11 different genetically defined subtypes of high-risk B lineage ALL by comparing leukemic cells of relapsed ALL patients with normal B cells obtained from human cord blood. The authors find that enrichment of accessible chromatin regions in leukemia. These ACRs are located in areas that do not contain histone modifications, which are characteristic of transcriptional activity in normal B cells. Indeed, most of the ACR in the leukemia cells also contained H3K27ac modifications suggesting the presence of transcriptional activity.

The authors further show that the different subtypes exhibited distinguishable Profiles of ACRs. Transcription factor motif analysis showed a certain correlation with the transcription of the respective target genes. The association of ACRs with leukemia related single nucleotide polymorphisms suggest that these ACRs might be related to activity of neighboring genes. However, transcriptional activity is directly documented in only two of such genes.

When comparing ACRs and transcriptional Profiles of samples obtained either at diagnosis or at the time of relapse the authors identify an association of a cluster ACRs and deregulated target genes with relapse free survival. Relapse-free patients thus separate from those experiencing a relapse, which mostly correlates with known genetic subtypes associated with favorable and unfavorable Prognosis, respectively. In this analysis, 3 hyperdiploid patients (a subgroup that is generally associated with a favorable risk Profile) who developed a relapse were correctly clustered together with the other high-risk patients who also developed a relapse. These leukemias were found to mimic an expression Profile of more immature cells and of myeloid Progenitors suggesting the presence of known high-risk features, which are not commonly associated with hyperdiploid precursor B-ALL. This expression Profile was also found in relapsing patients in another set of hyperdiploid patients documented in the TARGET database.

In sum, the data presented here describe features of chromatin accessibility in precursor B-ALL in comparison to a heterogenous population of normal B-cells. This part of the paper suffers from the poor definition of the normal cells. In particular, it is known that ALL cells are derived from immature precursors. Therefore, it is critical to compare leukemic cells with normal early B-cell

precursors. The reference B cells obtained from cord blood are likely a mixture of some immature and predominantly mature normal cells. Any differences between leukemic cells and such an ill-defined mixture of normal cells might thus represent differences between mature and immature cells and not necessarily between leukemic cells and their normal counterpart of comparable immaturity.

By contrast, the comparative data of diagnostic and relapse samples are potentially of more substantial interest. The identification of a high-risk expression and ACR Profile in hyperdiploid precursor B-ALL even at the time of initial diagnosis is novel and might be clinically useful as biomarker indicating risk. However, unequivocally documenting this usefulness would require the analysis of a larger clinically well annotated group of unselected patients rather than the 43 hyperdiploid samples extracted from the TARGET database.

[Author's response] We appreciated the insightful suggestions of the reviewer. There are 3 concerns raised by the reviewer, including the using of normal B cells from cord blood as control; the further validation of RFS associated ACR profile identified in B-ALL, especially in hyperdiploidy B-ALLs; the insufficient analysis of subtype enriched transcription factors. We responded to each concern in details below.

(1) Firstly, we agree with the reviewer the use of normal B-cells collected from cord blood would bias the interpreting of chromatin accessibility changes in B-ALL. In this revision, we re-analyzed the data to directly compare the chromatin accessibility of B-ALL to progenitor B cells. Publicly available ATAC-seq data were downloaded for pre-pro B and pro B cells (O'Byrne et al., 2019. PMID: 31383639) as a new reference. We observed comparable number of ACRs in individual genome between pre-pro B/pro B cells and B-ALLs, as showed in the revised Supplementary Figure 3B (attached below). We have modified the results on page 6 to represent this change in the revised manuscript.

Supplementary 3B (revised). Differences in accessible chromatin regions between B progenitor cells and B-ALLs. For each sample, data were down sampled to twenty million, thirty million and forty million reads for comparison. Wilcoxon test, $P = 0.5800$ for twenty million, $P = 0.9100$ for thirty million, $P = 0.4100$ for forty million.

We next compared the ACRs detected in B-ALLs to pre-pro B and pro B cells. Results showed that the majority of ACRs in pre-pro B cells (98.57%) and pro B cells (98.35%) were also present in B-ALLs (showed as a new Figure 1C), further supporting the B progenitor origin of B-ALL. Besides, we detected 585,248 ACRs showed specific accessibility in B-ALL, with high heterogeneity as compared to the ACRs shared between B-ALL and pre-pro B/pro B cells (showed in the new Figure 1D). We further identified a total of 252,028 recurrent ACRs with higher accessibility in B-ALL compared to B-cell progenitors. Enrichment analysis showed that these ACRs were associated with genes in tumor-associated biological processes including proliferation/differentiation, immune process, signal transduction and metabolic process (showed as a new Figure 1E and new Supplementary Table 14). Among these ACRs, we found increased chromatin accessibility in both the TSS regions of oncogenes (*IL7R*, *TCL1A*, *TCF3*, *RHOA* and *ELL*, among others, showed as a new Figure 1F) and known enhancer regions regulating transcription of oncogene (distal blood enhancer cluster (BENC) of *MYC*, showed as revised figure panel in Figure 1G). These observations suggested that chromatin accessibility changes were involved in B-ALL.

These results were organized in the “B-ALL-specific chromatin accessible regions associated with leukemogenesis” paragraph in the results section on pages 6 to 7 and revised Figure 1C-G (attached below) in the revised manuscript.

Figure 1C-G (revised). (C) Venn diagram shows the overlap between ACRs detected in B-ALL and B-cell progenitors (pre-pro B cells and pro B cells). (D) Violin plot presents the recurrence of ACRs in B-ALLs. The ACRs were grouped into four groups as showed in Figure 1C. Significant difference was observed among the four groups ($P < 0.0001$, Kruskal-Wallis test). (E) Gene set enrichment analysis shows that 2,332 protein coding genes regulated by 252,028 higher accessible ACRs in B-ALL were enriched in tumor associated biological processes (Supplementary Table 6). Only terms with $FDR < 0.001$ are displayed. The node size represents the enriched FDR values, and the edge represents overlap between two gene sets. Clusters of functionally related categories were manually grouped and labeled in different colors. (F) Wiggle plot shows regions with increased chromatin accessibility in B-ALLs as compared to B-cell progenitors. ACRs with +/- 1kb centered the TSS of representative cosmic genes are showed. Only subtypes with more than three cases are included and two samples are randomly selected and showed for each subtype. The ACR present higher accessible in B-ALL upstream TSS of *IL7R* are highlighted in light-yellow. (G) Wiggle plot shows the chromatin accessibility in the blood enhancer cluster (BENC) region (chr8:126,712,193 – 128,412,193). The positions of enhancers (A to I) are indicated in the BENC track

on the top. The tracks showing ACRs in this region are organized as in Figure 1F. Two enhancers from the BENC cluster showing increased accessibility in B-ALLs are highlighted in light-yellow.

Another analysis related to the use of normal B cells from cord blood in this analysis was annotation of ACRs detected in B-ALL to functional genomic regions. During the revision, we have replaced the functional partitioning of the genome with primary B-ALL as reference. ChIP-seq data of H3K27Ac, H3K4me1, H3K4me3, H3K9me3, H3K36me3, and H3K27me3 were downloaded from primary B-ALL patient (Blueprint Epigenomic Consortium) and a B-ALL cell line (Nalm6, data collected from PMID: 26219304 for GSE44218 and PMID: 30500954 for GSE109377). We re-constructed the genome partitioning with chromHMM in primary B-ALL and Nalm6 separately. We observed consistent results between primary B-ALL and Nalm6, with 97.1% of the genomic functional partitioning results were overlapped, as showed in the Extended Figure 4 below. This indicated the active and repressive partition of the genome was stable in the same cell origin. With this, we annotated the ACRs detected in B-ALLs with the updated genome partitioning annotation from primary B-ALL (included as a new Supplementary Figure 2B in the revised manuscript). Overall, the results were consistent with the observations in original manuscript. We showed that genome associated with potentially active transcription had higher chromatin accessibility compared to the repressive regions (Supplementary Figure 2C) and a median of 27.95% of the ACRs in each individual B-ALL sample were associated to Quies regions. We noticed that ACRs in Quies regions showed great heterogeneity among B-ALLs, indicating the observation of Quies ACRs could still be biased due to the unavailable of ChIP-seq data for individual tumor sample. Our analysis of H3K27ac ChIP-seq in 12 primary B-ALLs showed overlap between the H3K27ac and a subset of Quies ACRs, partially supporting this possibility.

The updates of this part were included in the results section on pages 5-6 under “Chromatin accessibility landscape of pediatric B-ALL” and also reflected in the revised discussion section on page 13. All the figures related to the functional genome partitioning were updated in the revised manuscript, including Figures 1-3, Supplementary Figures 2 and 4.

Extended Figure 4. Comparison of active and repressive genome partitioning in primary B-ALL (S017E3) and Nalm6. The genome partitioning was analyzed separately in primary B-ALL and Nalm6 with ChIP-seq data of H3K27Ac, H3K4me1, H3K4me3, H3K9me3, H3K36me3, and H3K27me3. Active regions include TssA, Tx, Enh and BivR. Repressive regions include ReprPC, Het, ZNF/Rpts and Quies.

(2) Secondly, we agree with the reviewer that the RFS associated ACRs and its potential value in distinguishing patients with early relapse need further validation in a larger unselected cohort. In the original analysis, we only explored the validation focusing on the hyperdiploidy subtype as patients with this subtype were considered as good prognosis, yet our analysis identified a subset of hyperdiploidy B-ALLs with the RFS associated ACR signature could relapse early. Following this thread, we searched for additional publicly available dataset with more hyperdiploidy B-ALL patients and prognosis information and examined with either ATAC-seq or RNA-seq. Unfortunately, we could not obtain such type of data to perform further validation. And this would be explored in future investigation. With this, we added the limitation of patient numbers on page 12.

Meanwhile, we performed a new analysis to expand the validation of the RFS associated ACRs to entire B-ALL cohort in TARGET project (252 samples were analyzed), including major B-ALL subtypes. To do this, we compared the ACRs between Group B (early relapse cases) and Group A from the clustering analysis in Figure 5B. We identified a total of 10,975 RFS associated ACRs with higher accessibility in B-ALLs from Group B ($|\log_2FC| > 1$ and $FDR < 0.05$) and found 1,827 potential target genes from ACR-to-gene association analysis (as a new

Supplementary Table 14). With these genes, 252 B-ALL samples from TARGET project were grouped into 3 clusters (as a new Supplementary Figure 7D and new Supplementary Table 15). Survival analysis showed significant difference in both event-free survival (EFS) and overall survival (OS) between the clusters (as a new Supplementary Figure 7E). Patients with the highest expression of the target genes showed the worst prognosis (Cluster 3). Noticeably, the expression signature was observed in the matched diagnosis tumors of the patients with inferior prognosis. For 12 out of 15 relapsed tumors in Cluster 3, the paired diagnosis tumors were also clustered in the same cluster with high-risk expression signature (as in new Supplementary Table 15). These results supported the chromatin accessibility changes were of potential value in predicting prognosis.

We have included this part of results in the revised Supplementary Figure 7D-E (attached below) and modified relevant writings on page 11 in the revised manuscript.

Supplementary Figure 7D-E (revised). (D) Unsupervised clustering analysis of 252 B-ALL samples from the TARGET project based on the expression of target genes potentially regulated by differential ACRs with increased accessibility in B-ALLs of Group B. (E) Event-free survival rate (EFS, 202 patients) and overall survival rate (OS, 201 patients) estimates for B-ALL samples of three clusters presented in Supplementary Figure 7D. Only patients with sufficient follow-up information were included. For patients with paired diagnosis and relapsed samples, the patient was assigned to the cluster of relapsed samples in the survival analysis. Log-rank test, EFS $P = 2.564e-28$ and OS $P = 5.038e-17$.

(3) Thirdly, we performed expression analysis on both the subtype enriched transcription factors (TF) and their potential target genes to explore more systematically of the subtype enriched TFs. As we replaced the normal B cells from cord blood with pre-pro B/pro B cells in the analysis during the revision, the subtype enriched TF analysis was also updated and there were 109 TFs in the revised manuscript with minimal changes (106 out of the 109 TFs were identified as subtype enriched TFs in both original and revised analysis). We clustered these TFs into 9 clusters based on their motif enrichment in each subtype, as showed in the revised Figure 2E, and performed the expression analysis based on this clustering result.

For each TF, statistical analysis was performed to test the expression difference between tumor samples of the enriched subgroup versus the others. A total of 14 TFs were found with significantly increased transcription itself in the enriched subtype, suggesting the transcription regulation directly associated with the TF expression (showed as a new figure panel of Figure 2F). Meanwhile, the transcription of target genes for each TF was also evaluated. We focused on the transcription factors with the binding motif in gene promoter region (TSS +/- 1kb), and these genes were analyzed as the targets for each transcription factor. The individually analyzed results of target genes were further combined to represent the regulatory function of the transcription factor in the enriched subtype. As showed in the new Figure 2G, a total of 13 out of the 53 transcription factors included in this analysis were found with significantly higher expression of their target genes, supporting the increased transcription regulation activity in the enriched subtype. We noticed that 12 out of 13 TFs in the target gene analysis did not show expression changes of the TFs themselves, indicating a context dependent transcription regulation among B-ALL subtypes.

We have revised the manuscript on page 8 to reflect these changes. The revised Figure 2 is also attached below.

Figure 2 (revised). Subtype-specific chromatin accessibility in B-ALL. (A) The t-distributed stochastic neighbor embedding (t-SNE) plot showing the clustering of 75 B-ALL samples based on recurrent c-ACRs with top 10% highest variance. (B) Heatmap of Pearson correlation coefficients shows the inter-sample correlation of chromatin accessibility based on all recurrent ACRs. (C) Heatmaps of Pearson correlation coefficients based on ACRs in Enh regions (left) and ACRs in BivR regions (right) showing the inter-sample similarity of chromatin accessibility. (D) The accessibility of 17,981 subtype-specific ACRs in 64 B-ALL samples is shown

in heatmap. The x axis presents 64 B-ALL samples and y axis displays subtype-specific ACRs. (E) Unsupervised clustering of 109 transcription factors (TF) based on their enrichment in each subtype, as labeled on the right of this plot. (F) Differential expression analysis of subtype enriched TFs in B-ALL. The TFs are colored according to the clusters in Figure 2E. For each TF, statistical analysis was performed to test the expression difference between B-ALLs of the enriched subgroup versus the others. Each dot represents an individual TF. Horizontal dashed line represents $P = 0.05$ and vertical dashed line represents fold change (FC) = 1.2. Gene symbols of TFs with $P < 0.05$ and $FC > 1.2$ are showed. (G) Expression analysis of target genes of subtype enriched TFs. Only TFs with binding motif in gene promoter region (TSS +/- 1kb) were included. Differential expression analysis was performed for each target gene between B-ALLs grouped upon the enriched subgroups of TF. The median fold change (FC) of all target genes regulated by the individual TF is showed on x axes. Each dot represents a group of target genes for an individual TF. Horizontal dashed line represents $P = 0.05$ and vertical dashed line represents fold change (FC) = 1.2. Gene symbols of TF with target genes satisfied $P < 0.05$ and $FC > 1.2$ are labeled.

**Reviewer #3, expertise in pediatric leukemia genomics and epigenomics
(Remarks to the Author):**

In the current manuscript, the authors Wang et al, have generated chromatin accessibility Profiles on 79 paediatric B-ALL tumours from a total of 61 patients. This was combined with WGS and RNA-seq for a subset of patents and H3K27Ac ChIP-seq on another subset of tumours. These patents represented 11 B-ALL subtypes. This is an interesting data set to have for the field, but the paper suffers from being overly descriptive and limited novelty. The paper would have benefitted from functional experiments to support some of the interesting findings.

Major comments:

1) **Reviewer3:** *I have some concerns on the strategy used to purify tumour cells, those with low tumour count % according to the Table S2. Some of these are in the range of 20-40% only and yet it appears that the cells were sorted based on CD19+/CD10+. Some patients have no Proportions. How can the authors be confident that their strategy was sufficient to purify leukaemia cells? Albeit at diagnosis/relapse it is likely that the burden is very high. Furthermore, whilst they show two representative flow cytometry plots - patient A442 does not exist in the Table S2?*

[Author's response] We thanks the reviewer for this comment. We applied flow-cytometry based purification of leukemia cells in this study to minimized potential noise introduced by including different types of cells in analyzing chromatin accessibilities. The 20-40% of tumor cells on columns C/D (for diagnosis sample) and H/I (for relapsed sample) in Table S2 represent the percentage of cells that were identified as tumor cells before purification as examined by morphological examination or flow-cytometry in clinical test. And the percentage of CD19/CD10 in this table represented the percentage of tumor cells that were positive with CD19 or CD10 in clinic tests before purification (columns E/F for diagnosis samples and columns J/K for relapsed samples). We have modified the header of Supplementary Table S2 and added the description accordingly in this table to make this clearer. As listed in the table, the percentage of CD19/CD10 positive tumor cells was high for B-ALLs, and we selected CD19 or CD10 marker to purify tumor cells accordingly. For the samples this information was not available, CD19 was used to purify B-ALL tumor cells considering CD19 is one of the most commonly expressed markers for B-ALL and was used for diagnosis and CAR T-cell therapy in clinic. Furthermore, none of these patients was treated with CAR

T-cells which excluded the potential CD19 loss for the B-ALL tumor cells during treatment. For patient A442, we are sorry for the typo we made in the original text of Supplementary Figure 1B. This patient should be A422. We have corrected the typo and double checked the entire manuscript.

2) **Reviewer3:** *Landscape assignment for ACR's. I have a major concern on the authors partitioning of the genome into functional regions using primary B-cells as a reference. The genomic regions and their assignment require H3K27Ac, H3K4me1, H3K4me3 and H3K27me3 - none of which were carried out on these samples and so the claims that ACR's are in "enhancer" or "Quiescent" regions are misplaced. Similarly, to say the repressed regions has less accessibility is misleading, as again - the repressed region is in the normal B-cell. It might not be repressed in B-ALL and the histone modifications are likely to be very different between the different subtypes of B-ALL at different regions. In the absence of the authors undertaking the appropriate ChIP-seq experiments, this analysis and conclusions are limited.*

[Author's response] We agree with the reviewer on this comment. As also pointed out by reviewers 1 and 2, the difference observed between B-ALL and normal B cells from cord blood might reflect difference in cell origin. In the revised manuscript, we have updated the functional partitioning of the genome in the analysis, replacing the normal B cell from cord blood with primary B-ALL cells. We collected publicly available ChIP-seq data of H3K27Ac, H3K4me1, H3K4me3, H3K9me3, H3K36me3, and H3K27me3 from primary B-ALL patient (Blueprint Epigenomic Consortium) and a B-ALL cell line (Nalm6, data collected from PMID: 26219304 for GSE44218 and PMID: 30500954 for GSE109377). We reconstructed the genome partitioning with chromHMM in primary B-ALL and Nalm6 separately and compared the genome partitioning results. Consistent results were observed between primary B-ALL and Nalm6, with 97.1% of the genomic functional partitioning results were overlapped, as showed in the Extended Figure 4 below. This indicated the active and repressive partition of the genome was stable in the same cell origin. With this, we redo the annotation of the ACRs with this updated genome partitioning annotation (included as a new figure panel in Supplementary Figure 2B in the revised manuscript). Results showed that genome associated with potentially active transcription had higher chromatin accessibility compared to the repressive regions (Supplementary Figure 2C) and a median of 27.95% of the ACRs in each individual B-ALL sample were associated to Quies regions, consistent with the observations in original

manuscript. Meanwhile, we noticed that the accessible chromatin in Quies regions showed great heterogeneity among B-ALLs, leading to the possibility that the observation of ACR in Quies regions might be due to the unavailability of ChIP-seq data for individual tumor sample. Our analysis of H3K27ac ChIP-seq in 12 primary B-ALLs indeed observed overlap between the H3K27ac and a subset of Quies ACRs, partially supporting this possibility.

We have revised the manuscript to include this possibility in the discussion section on page 13. We also re-organized the results on pages 5-6 under “Chromatin accessibility landscape of pediatric B-ALL” in the results section. All the figures related to the functional genome partitioning were updated in the revised manuscript, including Figures 1-3, Supplementary Figures 2 and 4.

Extended Figure 4. Comparison of active and repressive genome partitioning in primary B-ALL (S017E3) and Nalm6. The genome partitioning was analyzed separately in primary B-ALL and Nalm6 with ChIP-seq data of H3K27Ac, H3K4me1, H3K4me3, H3K9me3, H3K36me3, and H3K27me3. Active regions include TssA, Tx, Enh and BivR. Repressive regions include ReprPC, Het, ZNF/Rpts and Quies.

3). **Reviewer3:** *Transcription factor motif analysis: The authors have analysed the ACR's and clustered them according to a predefined set of 110 different transcription factors. There is some nice TF sub-type specificity. However, the authors then seem to cherry pick E2F6 and THAP1 to show the relevance of this analysis and ignore all the others in the list that are enriched for each subtype. Can the authors look more globally at each motif and determine which TF's might*

be relevant based on expression? Moreover, they suggest overall between MLL rearranged and ZNF-384 subtypes, and yet there is greater overlap between E2A and ETV subtypes which they do not describe. Can the authors comment? The authors also claim in the figure 2 legend for panel E that this clustering is based on occupancies of the TF's. This is not the case - this is motif enrichment as no ChIP-seq was undertaken for all 110 TF's to support this claim.

[Author's response] We thank the reviewer for this suggestion. In the revised manuscript, we performed expression analysis on both the subtype enriched transcription factors (TF) and their potential target genes. As we replaced the normal B cells from cord blood with pre-pro B/pro B cells in the analysis during the revision, the subtype enriched TF analysis was also updated and there were 109 TFs in the revised results with minimal changes (106 out of the 109 TFs were identified as subtype enriched TFs in both original and revised analysis). We clustered these TFs into 9 clusters based on their motif enrichment in each subtype, as showed in the revised Figure 2E, and performed the expression analysis based on this clustering result.

For each TF, statistical analysis was performed to test the expression difference between tumor samples of the enriched subgroup versus the others. A total of 14 TFs were found with significantly increased transcription itself in the enriched subtype, suggesting the transcription regulation directly associated with the TF expression (showed as a new figure panel of Figure 2F). Meanwhile, the transcription of target genes for each TF was also evaluated. We focused on the transcription factors with the binding motif in gene promoter region (TSS +/- 1kb), and these genes were analyzed as the targets for each transcription factor. The individually analyzed results of target genes were further combined to represent the regulatory function of the transcription factor in the enriched subtype. As showed in the new Figure 2G, a total of 13 out of the 53 transcription factors included in this analysis were found with significantly higher expression of their target genes, supporting the increased transcription regulation activity in the enriched subtype. We noticed that 12 out of 13 TFs in the target gene analysis did not show expression changes of the TFs themselves, indicating a context dependent transcription regulation among B-ALL subtypes.

As the reviewer pointed out, besides the TFs enriched for a specific B-ALL subtype, we observed high similarity of TF enrichment between some of the B-ALL subtypes. This included shared TFs in TCF3::PBX1 and ETV6::RUNX1 subtypes, KMT2A and ZNF384 subtypes, and in BCR::ABL1/BCR::ABL1-like and hyperdiploidy subtypes, indicating the similarity in transcription regulation between the overlapped subtypes. We have revised the manuscript on page 8 to

reflect these changes. We also thank for the reviewer to point out the miss use of 'occupancy'. We have replaced the word with 'enrichment' in the revised manuscript. The revised Figure 2 is also attached below.

Figure 2 (revised). Subtype-specific chromatin accessibility in B-ALL. (A) The t-distributed stochastic neighbor embedding (t-SNE) plot showing the clustering of 75 B-ALL samples based on recurrent c-ACRs with top 10% highest variance. (B)

Heatmap of Pearson correlation coefficients shows the inter-sample correlation of chromatin accessibility based on all recurrent ACRs. (C) Heatmaps of Pearson correlation coefficients based on ACRs in Enh regions (left) and ACRs in BivR regions (right) showing the inter-sample similarity of chromatin accessibility. (D) The accessibility of 17,981 subtype-specific ACRs in 64 B-ALL samples were shown in heatmap. The x axis presents 64 B-ALL samples and y axis displays subtype-specific ACRs. (E) Unsupervised clustering of 109 transcription factors (TF) based on their enrichment in each subtype, as labeled on the right of this plot. (F) Differential expression analysis of subtype enriched TFs in B-ALL. The TFs are colored according to the clusters in Figure 2E. For each TF, statistical analysis was performed to test the expression difference between B-ALLs of the enriched subgroup versus the others. Each dot represents an individual TF. Horizontal dashed line represents $P = 0.05$ and vertical dashed line represents fold change (FC) = 1.2. Gene symbols of TFs with $P < 0.05$ and $FC > 1.2$ are showed. (G) Expression analysis of target genes of subtype enriched TFs. Only TFs with binding motif in gene promoter region (TSS +/- 1kb) were included. Differential expression analysis was performed for each target gene between B-ALLs grouped upon the enriched subgroups of TF. The median fold change (FC) of all target genes regulated by the individual TF is showed on x axes. Each dot represents a group of target genes for an individual TF. Horizontal dashed line represents $P = 0.05$ and vertical dashed line represents fold change (FC) = 1.2. Gene symbols of TF with target genes satisfied $P < 0.05$ and $FC > 1.2$ are labeled.

4) Reviewer3: *Analysis of Diagnosis and Relapse: Relapse remains an important and significant clinical challenge and epigenetic and genetic changes have been implicated. The authors integrate drug susceptibility with 1259 genes that are potentially deregulated in relapse. There is an association between Dasatinib and PH/PHL B-ALL. The authors then claim that treatment could reshape the chromatin. However, there may have also been clonal selection. Have the authors looked at clonal composition and outgrowth of clones with different mutations?*

[Author's response] We thanks the reviewer for this comment. Clonal evolution is commonly observed in relapse of leukemia. As reported in our previous study (Li et al., 2020. PMID: 31697823), almost all relapsed ALL patients experienced branched evolution. For the 7 BCR::ABL1/BCR::ABL1-like B-ALLs analyzed in current study, 6 had whole genome sequencing data available from previous analysis. Clonal evolution analysis showed that the relapse was seeded by single

dominant clone in 4 out of 6 patients, while in the other 2 patients the relapse was seeded by multiple subclones. The Extended Figure 5 below (adopted from previous study, Li et al., 2020. PMID: 31697823) showed clonal evolution pattern of the 6 BCR::ABL1/BCR::ABL1-like B-ALLs included in current analysis. In all these cases, there were subclones in diagnosis sample swept from relapse and additional relapse-specific mutations acquired. We agree with the reviewer that the selection of clones might contribute to the difference of chromatin accessibility observed between diagnosis and relapse tumors, as the difference of ACR might exist between subclones. Ideally, simultaneously analyze the chromatin accessibility and gene mutations at single cell level for paired diagnosis and relapse tumors would provide valuable information for this question. Unfortunately, we were not able to perform this analysis at this stage, due to the unavailability of high-quality cells. We have modified the writings on page 14 of the revised manuscript to include the potential clonal selection role in the different ACRs between diagnosis and relapse B-ALL.

Extended Figure 5. Two dimensional VAF plots of 6 BCR::ABL1/BCR::ABL1-like B-ALLs analyzed in current analysis showing the potential clonal evolution schemes. The plots were adopted from supplementary notes of previously published study (Li et al., 2020. PMID: 31697823).

Reviewers' Comments:

Reviewer #1:

Remarks to the Author:

Thank you very much for clarifications and for addressing many of the points raised. Overall, the manuscript in its revised form presents a valuable resource describing the accessible chromatin landscape in relapsed B-ALL, its molecular subtypes and relapse-specific phenotype. The sample size remains a major limitation for outcome analyses. Several remaining points are:

1.) The newly included comparison of B-ALL ATAC-Seq data to normal B precursor data provides an improved definition of B-ALL specific ACRs. It seems somewhat surprising that comparable ACRs per sample in normal B precursor and B-ALL ATAC-Seq samples (Figure S3), result in very distinct numbers of ACRs specific to B-ALL or normal B precursors in Figure 1C. This would need further clarification. Is this an effect of the distinct sample sizes? Then the representation would need to be adjusted for this.

Also, there are pre-published data available from a very similar analysis which used the same references for B-ALL to normal B precursor comparisons (<https://www.biorxiv.org/content/10.1101/2023.02.14.528493v1>). A discussion of the results would put the findings of this manuscript into context. A direct comparison of the data would help to appreciate core ACR sets specific to B-ALL and selected B-ALL subtypes across cohorts and would strengthen the identification of relapse-specific ACRs in the current manuscript. One limitation the new reference is the fetal origin of the normal B precursor samples. This should at least be mentioned in the corresponding paragraph of the main manuscript.

2.) The molecular subtype definition of the patient data set seems questionable. The authors state in the main manuscript that n=20 patients were allocated to the 'high hyperdiploid' ALL subtype (lines 104 f.). Later on, 'hyperdiploid' is used without clarifying if this refers to 'high hyperdiploid' cases or a distinct definition. According to Supplementary Table 1, out of n=20 'high hyperdiploid' cases, n=6 harbor either a normal karyotype or only minor chromosomal aberrations. This also refers to the karyotype of patient 233 (provided in the rebuttal) which to the best of our knowledge would not fit any definition of high hyperdiploid ALL. Patient 150, which is labeled 'Hypodiploidy' in Supplementary Table 1 also harbors a normal karyotype according to the data presented. It remains unclear how the ETV6::RUNX1 subtype was established in patients 451 and 414. - Please clarify! Please provide an overview of virtual karyotypes from WGS and describe how molecular subtypes were allocated in the named cases. Definitions should be then used throughout the manuscript.

3.) Based on the data presented, it remains difficult to conclude if the definitions of two proposed prognostic groups (Group A and B, Figure 5) are independent of the molecular disease subtype. Except for three hyperdiploid cases (two high hyperdiploids and one low hyperdiploid probably), groups represent molecular subtypes associated with prognosis. The authors argue that the overlap between RFS-related ACRs and subtype-specific ACRs as well as recurrent c-ACRs is comparatively small, which supports the assumption that the definition of prognostic groups is independent of the molecular subtype. However, the quantity of features included in the analysis is not necessarily reflecting their importance for clustering. A small number of subtype-specific ACRs could still be the major determinant of group separation. An analysis of the top discriminating ACRs between Group A and B and their relation to the molecular B-ALL subtypes would be more informative. Are RFS-specific ACRs shared across subtypes? What was the outcome of the three group B 'hyperdiploid' cases compared to the group A 'hyperdiploid' cases? Is this consistent with the TARGET data? Please specify which patient data set of the TARGET project was used? Was there also a selection for or enrichment of relapsed cases?

Reviewer #2:

Remarks to the Author:

The major points that I suggested to be addressed by the authors were

1. Poor definition of the normal cells. The authors have now included in the manuscript the analysis of publicly available ATAC–data of precursor B-cells. This reanalysis has resulted in a new list of genomic regions with differential chromatin accessibility in normal precursors and in precursor B-ALL cells. The authors have thus addressed this point of criticism in their revision.
2. As previously pointed out, the identification of high-risk patients within the subgroup of hyperdiploid precursor B-ALLs at the time of diagnosis could be potentially clinically useful. The authors have now tried to address this point by including more patients from the TARGET database. However, considering that hyperdiploid precursor B-ALL's are a common subgroup of this type of leukemia direct comparisons of a larger number of patients would have been preferable and would have excluded technical variabilities from different laboratories. Further, the clinical annotation of patients included in the TARGET database is incomplete limiting the potential for clinical meaningful analysis. Therefore, the data provided here remain descriptive without much clinical usefulness.

Reviewer #3:

Remarks to the Author:

I would like to commend the authors for addressing all my comments, which have strengthened the manuscript. I have no further comments and look forward to seeing this important data in the public domain.

Point-by-point Response to Reviewers' Comments

Reviewer #1 (Remarks to the Author):

Thank you very much for clarifications and for addressing many of the points raised. Overall, the manuscript in its revised form presents a valuable resource describing the accessible chromatin landscape in relapsed B-ALL, its molecular subtypes and relapse-specific phenotype. The sample size remains a major limitation for outcome analyses. Several remaining points are:

*1.) **Reviewer1:** The newly included comparison of B-ALL ATAC-Seq data to normal B precursor data provides an improved definition of B-ALL specific ACRs. It seems somewhat surprising that comparable ACRs per sample in normal B precursor and B-ALL ATAC-Seq samples (Figure S3), result in very distinct numbers of ACRs specific to B-ALL or normal B precursors in Figure 1C. This would need further clarification. Is this an effect of the distinct sample sizes? Then the representation would need to be adjusted for this.*

[Author's Response] Thanks for this comment. As the reviewer pointed out, although a comparable number of ACRs were detected in normal pre-pro B/pro B cells and B-ALLs, we observed that a large proportion of ACRs (78.39%) were B-ALL specific (Figure 1C). On the other hand, majority of ACRs detected in pre-pro B (98.57%) and pro B (98.35%) cells were also detected in B-ALLs (Figure 1C). This observation is because of the heterogeneity of ACRs in B-ALLs. As we showed in Figure 1D, only 69.52% of the B-ALL specific ACRs were recurrent (detected in more than 2 cases) across the B-ALLs, compared to 99.80% in ACRs overlapped between B-ALL, pre-pro B and pro B cells. We described this potential explanation on page 6 in the previous revision.

Reviewer1: *Also, there are pre-published data available from a very similar analysis which used the same references for B-ALL to normal B precursor comparisons (<https://www.biorxiv.org/content/10.1101/2023.02.14.528493v1>). A discussion of the results would put the findings of this manuscript into context. A direct comparison of the data would help to appreciate core ACR sets specific to B-ALL and selected B-ALL subtypes across cohorts and would strengthen the identification of relapse-specific ACRs in the current manuscript.*

[Author's Response] We have included this pre-published study in the discussion section on page 15 during this revision. We will follow this study and data to perform an integrated analysis in the future.

Reviewer1: *One limitation the new reference is the fetal origin of the normal B precursor samples. This should at least be mentioned in the corresponding paragraph of the main manuscript.*

[Author's Response] We revised the results on page 6 to state that the normal B-cell precursors included in current analysis were collected from fetal bone marrow in this revision.

2.) **Reviewer1:** *The molecular subtype definition of the patient data set seems questionable. The authors state in the main manuscript that n=20 patients were allocated to the 'high hyperdiploid' ALL subtype (lines 104 f.). Later on, 'hyperdiploid' is used without clarifying if this refers to 'high hyperdiploid' cases or a distinct definition. According to Supplementary Table 1, out of n=20 'high hyperdiploid' cases, n=6 harbor either a normal karyotype or only minor chromosomal aberrations. This also refers to the karyotype of patient 233 (provided in the rebuttal) which to the best of our knowledge would not fit any definition of high hyperdiploid ALL. Patient 150, which is labeled 'Hypodiploidy' in Supplementary Table 1 also harbors a normal karyotype according to the data presented. It remains unclear how the ETV6::RUNX1 subtype was established in patients 451 and 414. - Please clarify! Please provide an overview of virtual karyotypes from WGS and describe how molecular subtype were allocated in the named cases. Definitions should be then used throughout the manuscript.*

[Author's Response] We thank the reviewer for this comment. The 'hyperdiploidy' refers to the same subtype as 'high hyperdiploidy'. We have modified the manuscript in this revision with 'hyperdiploidy'.

The molecular subtypes of B-ALL were classified by combining the results of following analysis: (1) gene expression pattern-based subtype classification by our in-house developed recurrent neural network (RNN) based model (Cui B. et al, manuscript in preparation), (2) fusions, structure variations and driver mutations detected in RNA-seq and/or whole genome sequencing data, (3) the CNV results from whole genome sequencing and RNA-seq analysis, (4) karyotyping from clinical test. For each individual case, results from all above-mentioned analyses

were collected and manually curated for subtype classification. The resulted molecular subtype will be cross validated for the cases with both diagnosis and relapsed samples analyzed. Cases that could not be classified by this process were grouped into B-other. We included the results of these analyses in the revised Supplementary Table S1 and updated the Methods section accordingly with the above described molecular subtype classification methods in this revision.

Specifically for the cases mentioned by the reviewer, Patient 451 and 414 showed gene expression profiles representing ETV6::RUNX1 subtype. And ETV6::RUNX1 fusion were detected in tumor samples of both patients (Supplementary Table S1). These two cases were grouped as ETV6::RUNX1 subtype. Patient 150 did not have a high confidence subtype prediction based on gene expression analysis. CNV analysis of RNA-seq data showed one copy loss of 10 chromosomes in both diagnosis and relapsed tumor sample of this case, including chromosome 2, 3, 4, 7, 9, 12, 13, 16, 17 and 20. Consistent CNV results were observed in WGS as showed in our previous report (Li et al. Blood. 2020 Jan 2;135(1):41-55.). This case was grouped as hypodiploidy subtype. Patient 233 showed gene expression profile representing hyperdiploidy B-ALL and was detected copy number gains on chromosome 17, 21 and X. While no karyotyping result available for this case, it was grouped into hyperdiploidy subtype based on above results.

The molecular subtype results were further validated by clustering the current cohort with TARGET B-ALLs. We downloaded the RNA-seq data of 105 B-ALL samples with detailed subtype information from TARGET database (phs000464 from dbGap). As shown in the t-SNE plot below in Extended Figure 1, the 89 samples from 57 B-ALL patients in our studies were grouped together with the B-ALLs from TARGET cohort of the same subtype, as labeled with different colors in this plot, supporting the molecular subtype classification of B-ALLs in current analysis.

Extended Figure 1. The t-distributed stochastic neighbor embedding (t-SNE) plot showing the clustering of 105 B-ALL samples with confirmed molecular subtype of TARGET and 89 B-ALL samples from our study. The shape of each dot presents study cohort, and colors represents the molecular subtypes. The samples of four patients mentioned by the reviewer were highlighted with black box.

3.) **Reviewer1:** *Based on the data presented, it remains difficult to conclude if the definitions of two proposed prognostic groups (Group A and B, Figure 5) are independent of the molecular disease subtype. Except for three hyperdiploid cases (two high hyperdiploids and one low hyperdiploid probably), groups represent molecular subtypes associated with prognosis. The authors argue that the overlap between RFS-related ACRS and subtype-specific ACRs as well as recurrent c-ACRs is comparatively small, which supports the assumption that the definition of prognostic groups is independent of the molecular subtype. However, the quantity of features included in the analysis is not necessarily reflecting their importance for clustering. A small number of subtype-specific ACRs could still be the major determinant of group separation. An analysis of the top discriminating ACRs between Group A and B and their relation to the molecular B-ALL subtypes would be more informative. Are RFS-specific ACRs share across subtypes?*

[Author's Response] We analyzed the ACRs with top 10% most significant FDR which represent the top discriminating ACRs between Group A and B. These ACRs were the same as used in the t-SNE plot in Figure 5B separating Group A and Group B. Among these 7,059 ACRs, 1,078 (15.27%) ACRs were only accessible in single subtype, as showed in the Extended Figure 2A below. Similar result was observed when analyzing all the 70,592 RFS-related ACRs, 10,363 (13.10%) ACRs were found in cases of only one subtype (Extended Figure 2B). These results showed that the RFS associated ACRs were largely shared between B-ALL subtypes.

Extended Figure 2. Upset plot showing the RFS associated ACRs across subtypes. (A) The intersections of 7,059 RFS-related ACRs with the top 10% most significant FDR value across subtypes. The ACRs were the same as used in the t-SNE presentation in Figure 5B. The columns in red were ACRs shared in at least two subtypes, and the columns in blue present for ACRs accessible only in one subtype. The number of ACRs in each category was listed on top of the columns. (B) The intersections of all 70,593 RFS-related ACRs across subtypes. The data was

presented as described in 2A.

Reviewer1: *What was the outcome of the three group B ‘hyperdiploid’ cases compared to the group A ‘hyperdiploid’ cases? Is this consistent with the TARGET data? Please specify which patient data set of the TARGET project was used? Was there also a selection for or enrichment of relapsed cases?*

[Author’s Response] Two of three hyperdiploidy cases in Group B didn’t have the overall survival information, so we only performed RFS analysis during this revision. The three hyperdiploidy cases in Group B showed significantly worse prognosis as compared to cases in Group A (Extended Figure 3 below), consistent with the result from analyzing TARGET cohort as showed in Figure 5F and 5G. As we stated in the data availability section of the manuscript, the TARGET dataset used in this analysis was collected from dbGaP with sub-study ID phs000464. This is a high-risk B-ALL cohort as described in the TARGET project webpage and dbGap. All 43 samples with definitive molecular evidence for hyperdiploidy subtype in this dataset were included in our analysis without further selection.

Extended Figure 3. The Relapse-free survival rate (RFS) analysis between hyperdiploidy cases of Group A and Group B. (Log-rank test, P = 0.003433)

Reviewer #2 (Remarks to the Author):

Reviewer2: *The major points that I suggested to be addressed by the authors were*

1. *Poor definition of the normal cells. The authors have now included in the manuscript the analysis of publicly available ATAC–data of precursor B-cells. This reanalysis has resulted in a new list of genomic regions with differential chromatin accessibility in normal precursors and in precursor B-ALL cells. The authors have thus addressed this point of criticism in their revision.*

2. *As previously pointed out, the identification of high-risk patients within the subgroup of hyperdiploid precursor B-ALLs at the time of diagnosis could be potentially clinically useful. The authors have now tried to address this point by including more patients from the TARGET database. However, considering that hyperdiploid precursor B-ALL's are a common subgroup of this type of leukemia direct comparisons of a larger number of patients would have been preferable and would have excluded technical variabilities from different laboratories. Further, the clinical annotation of patients included in the TARGET database is incomplete limiting the potential for clinical meaningful analysis. Therefore, the data provided here remain descriptive without much clinical usefulness.*

[Author's Response] We thank the reviewer for the suggestions about the comparison B-ALL and normal B cells. We will keep following and analyzing of the pattern observed in the hyperdiploidy cases with inferior prognosis in the future.

Reviewer #3 (Remarks to the Author):

Reviewer3: *I would like to commend the authors for addressing all my comments, which have strengthened the manuscript. I have no further comments and look forward to seeing this important data in the public domain.*

[Author's Response] Thank you for the informative suggestions to improve our manuscript.

Reviewers' Comments:

Reviewer #1:

Remarks to the Author:

I would like to thank the authors for the further clarifications, which fully address most of my points. However, sample allocation to the High hyperdiploid subtype remains one last critical point. We fully agree with the authors, that in some instances conventional and virtual karyotypes and gene-expression-based subtype definitions are not in full agreement, requiring manual curation. However, the authors might agree that n=6/20 cases classified as 'high hyperdiploid' despite nearly diploid or hypodiploid chromosome counts by conventional karyotyping is an unexpected high number. – Especially, since virtual karyotyping from RNA-Seq data seems to confirm near diploid karyotypes in 3 cases. Case 472 is possibly rather a doubled near haploid karyotype which was not detectable by conventional karyotyping? The authors should at least clarify this discrepancy between gene-expression-based subtype allocation and conventional / virtual karyotyping in the main manuscript. A per-case depiction on why the questionable cases were classified in that way would be very helpful. Use of publicly available tools for gene-expression-based subtype allocation (e.g. PMIDs: 35482550, 35562965, 37645423) as an independent validation would strengthen the corresponding subtype allocations.

Point-by-point Response to Reviewers' Comments

Reviewer #1 (Remarks to the Author):

I would like to thank the authors for the further clarifications, which fully address most of my points. However, sample allocation to the High hyperdiploid subtype remains one last critical point. We fully agree with the authors, that in some instances conventional and virtual karyotypes and gene-expression-based subtype definitions are not in full agreement, requiring manual curation. However, the authors might agree that n=6/20 cases classified as 'high hyperdiploid' despite nearly diploid or hypodiploid chromosome counts by conventional karyotyping is an unexpected high number. – Especially, since virtual karyotyping from RNA-Seq data seems to confirm near diploid karyotypes in 3 cases. Case 472 is possibly rather a doubled near haploid karyotype which was not detectable by conventional karyotyping? The authors should at least clarify this discrepancy between gene-expression-based subtype allocation and conventional / virtual karyotyping in the main manuscript. A per-case depiction on why the questionable cases were classified in that way would be very helpful. Use of publicly available tools for gene-expression-based subtype allocation (e.g. PMIDs: 35482550, 35562965, 37645423) as an independent validation would strengthen the corresponding subtype allocations.

[Author's Response] Thank you for this question. As we described in the last revision of our manuscript, the B-ALL subtypes was classified by combining the results of gene expression pattern-based subtype prediction; detection of fusions, structure variations, driver mutations and CNV aberrations from RNA-seq and/or whole genome sequencing data; and karyotyping from clinical test. This strategy is commonly applied in the genomics analysis of pediatric B-ALL. One most recent example is in Samuel B. et al (Nature Genetics 2022) with 2,754 B-ALLs analyzed. As the reviewer mentioned, results from different analysis could be inconsistent in some cases. This is because of the complexity of the genomic aberrations of this disease. And in this case, we need to make the subtype allocation to best represent the molecular signatures of the case analyzed. In some case, there will be primary and secondary subtype allocated for a single case, as showed in the recent B-ALL genomic landscape paper (Samuel B. et al, Nature Genetics 2022).

For the 6 hyperdiploidy cases with normal karyotyping results collected from clinic test, chromosome gains were observed for 3 cases with the CNV analysis from

RNA-seq data (213, 350 and 496). We further carried out the subtype prediction analysis with the tools suggested by the reviewer, namely ALLSort (PMID 35482550), Allspice (PMID 35562965) and ALLCatchR (PMID 37645423). High confident prediction of hyperdiploidy subtype were predicted for all these 3 cases with at least one of the tools for each case (Allspice and ALLCatchR for 213; ALLCatchR for 350; ALLSort, Allspice and ALLCatchR for 496), supporting the subtype classification of these cases. For the rest 3 cases (357, 409 and 472), none of the tools could give high confident result. And as the reviewer pointed out, case 472 is one most difficult case and might be debatable as inconsistent results from karyotyping and CNV analysis were observed. The major evidence to classify these cases was based on the unsupervised clustering analysis based on gene expressions and the prediction of our in-house RNN based classification tool. All these 3 cases were clustered with other hyperdiploidy B-ALLs. As we showed in our previous revision, this cluster result could be further supported when the B-ALL from TARGET project was included. We believe this analysis combining the B-ALLs from an independent cohort with the subtype classified completely independently could serve as a strong validation of our result. Furthermore, this could also be supported by the unsupervised clustering analysis from chromatin accessibility results. As we showed in Figure 2A of our manuscript, all these cases were clustered tightly with other hyperdiploidy cases. We also attach below this Figure 2A with the case names labeled to make this clearer. This result serves as another independent validation of our subtype classification. Taking together, we believe the B-ALL subtype classification in our analysis represent the current understanding of this malignancy and the conclusions we draw from this analysis are robust.